# The $Sh3Pxd2b^{nee−/−}$ mouse reveals developmental features of Frank-ter Haar syndrome

Julika Huber[1,2,*], Siddharth Menon[1,3], Michael Lopez-Torres[1], Jason L. Guo[1] and Michael T. Longaker[1,3]

## ABSTRACT

Frank-ter Haar syndrome (FTHS) is an inherited disease associated with variants of the *SH3PXD2B* gene, encoding for the podosomal adaptor protein known as TKS4. FTHS is characterized by multiple skeletal abnormalities, developmental delay and severe craniofacial dysmorphology. This study provides an in-depth characterization of the calvarial phenotype of a mouse model of FTHS and investigates the potential underlying molecular and transcriptomic mechanisms. The $Sh3Pxd2b^{nee−/−}$ mouse presents with craniofacial malformations and disrupted suture patterning, as well as reduced osteoregeneration and decreased cell proliferation and migration observed both *in vitro* and *in vivo*, and impaired podosome formation. Transcriptomic analysis revealed downregulation of genes involved in ribosome biogenesis. Moreover, ribosomal RNA accumulates in cell protrusions of migrating cells. We established that the craniofacial phenotype of the $Sh3Pxd2b^{nee−/−}$ mouse is governed by impaired cell migration and proliferation due to dysfunctional podosome formation, particularly in neural crest-derived tissues. Transcriptomic and molecular data suggest altered ribosome-related processes, although further investigation is needed to clarify the underlying mechanisms.

KEY WORDS: Frank-ter Haar syndrome, Podosome, Craniofacial development, Suture patterning, Neural crest cell migration, Ribosome biogenesis

## INTRODUCTION

The mammalian skull vault comprises five membranous flat bones with an overlying periosteum and an underlying dura mater (Morriss-Kay, 2001; Opperman, 2000). The paired frontal bones and paired parietal bones are connected by the coronal suture. The parietal bones are connected to the unpaired interparietal bone by the lamboid suture. Along the midline, frontal and parietal bones are separated by the posterofrontal (PF) suture and the sagittal (Sag) suture, respectively. The anterior and posterior fontanel represent sites of confluence of more than two calvarial bones (Grova et al., 2012; Lenton et al., 2005).

The head mesenchyme originates from two main sources with the parietal bones deriving from paraxial mesoderm, the frontal bones, the PF suture and the dura mater deriving from the neural crest, and the Sag and coronal suture with mixed origins (Morriss-Kay, 2001; Jiang et al., 2002; Lenton et al., 2005). Cranial sutures are the leading sites of bone growth in craniofacial development.

Podosomes, actin-rich protrusions on the cell membrane, contribute to embryo development by facilitating cell migration, extracellular matrix degradation and cell adhesion (Dülk et al., 2016; Buschman et al., 2009; Courtneidge, 2012; Yang et al., 2011; Cejudo-Martin and Courtneidge, 2011; Murphy and Courtneidge, 2011; Alonso et al., 2020). These small organelles act as adhesion sites between the actin skeleton and the extracellular matrix, constantly assembling and disassembling (Alonso et al., 2020). Multiple human genetic syndromes have been associated with variants of podosomal proteins leading to alterations in craniofacial development (Iqbal et al., 2010; Cejudo-Martin and Courtneidge, 2011).

The rare, autosomal recessive disease Frank-ter Haar syndrome (FTHS) causes severe craniofacial dysmorphology and skeletal malformations, eye and cardiac abnormalities (Kudlik et al., 2020; Dülk et al., 2016; Vas et al., 2019; Bendon et al., 2012; Iqbal et al., 2010). Variants of the *SH3PXD2B* gene are responsible for FTHS. In humans, the *SH3PXD2B* gene, mapping to chromosome 5q35, encodes for the adaptor protein known as TKS4. TKS4 regulates several cellular processes, such as extracellular matrix degradation to promote cell migration and trafficking, epidermal growth factor receptor signaling, production of reactive oxygen species, formation of podosomes and invadopodia, and mesenchymal stem cell differentiation (Kudlik et al., 2020; Dülk et al., 2016; Vas et al., 2019).

Using the $Sh3Pxd2b^{nee−/−}$ mouse as a model for FTHS, we carried out in-depth characterization of skull development and gained insights into the potential molecular and transcriptomic mechanisms governing the craniofacial phenotype of FTHS.

## RESULTS

### $Sh3Pxd2b^{nee−/−}$ mice are characterized by severe craniofacial dysmorphology and disrupted suture patterning

The *SH3PXD2B* gene encodes for TKS4, a member of the tyrosine kinase substrate family of SRC kinases, which contains one Phox homology (PX) domain and four SRC homology 3 (SH3) domains. The $Sh3Pxd2b^{nee−/−}$ mouse harbors a spontaneous single base pair deletion in the last exon of the *Sh3pxd2b* gene on chromosome 11. This frame-shift mutation causes a protein truncation, which alters a part of the third SH3 and deletes the fourth SH3 domain (Mao et al., 2009). Owing to the type of mutation in the $Sh3Pxd2b^{nee−/−}$ mouse, the level of TKS4 protein should be unaffected compared to wild-type (WT) mice, but the protein is not fully functional.

$Sh3Pxd2b^{nee−/−}$ mice are characterized by a smaller skeleton and a shortened nose and skull (Fig. 1A). Micro-computed tomography (micro-CT) images (Fig. 1B) of the $Sh3Pxd2b^{nee−/−}$ mouse

[1]Department of Surgery, Division of Plastic and Reconstructive Surgery, Stanford University School of Medicine, Stanford, CA 94305, USA. [2]Department of Plastic Surgery, BG University Hospital Bergmannsheil, Ruhr University Bochum, 44789 Bochum, Germany. [3]Stanford Institute for Stem Cell Biology and Regenerative Medicine, Stanford University School of Medicine, Stanford, CA 94305, USA.

*Author for correspondence ( jlhuber@stanford.edu)

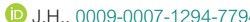 J.H., 0009-0007-1294-7794

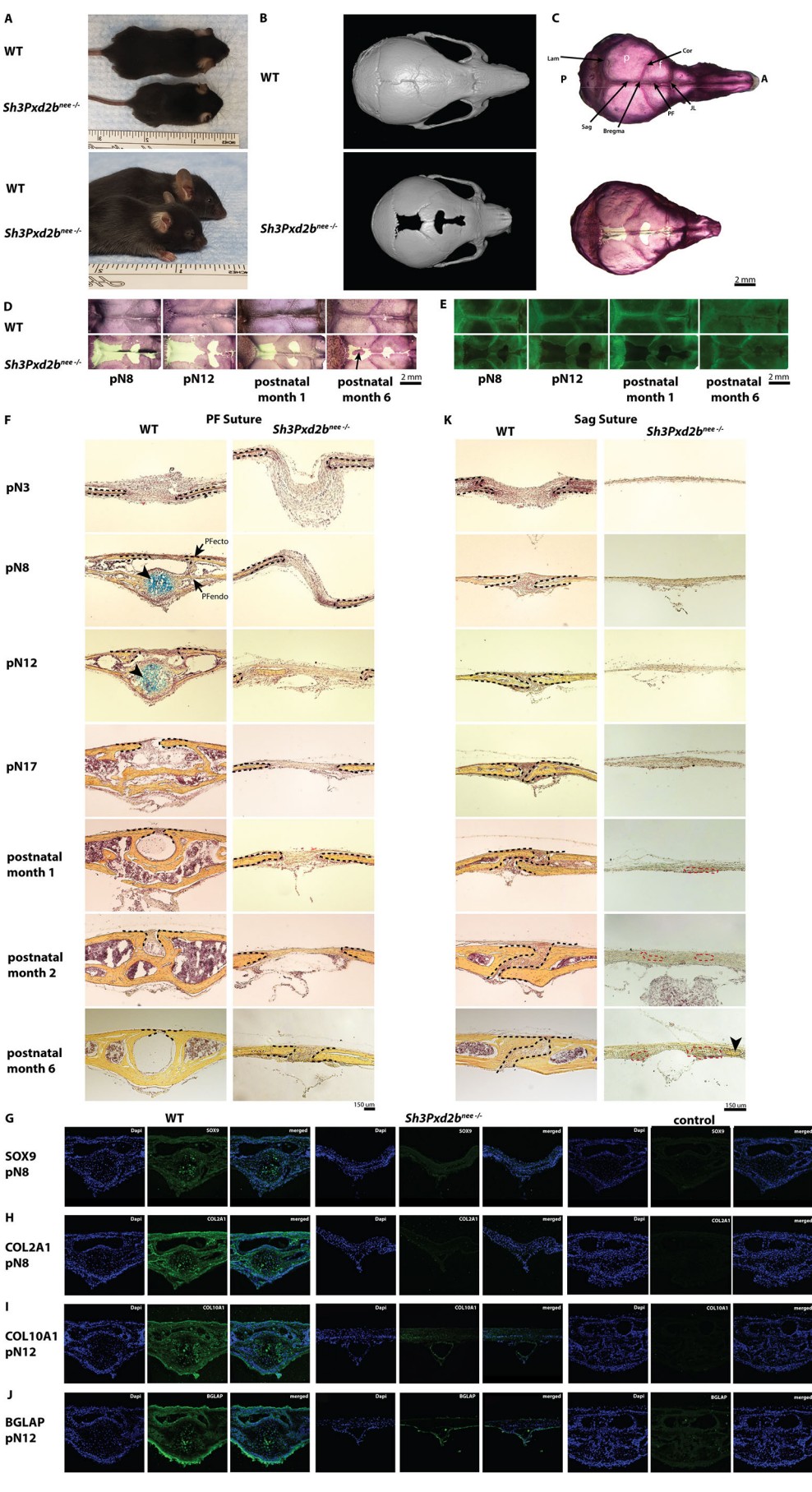

**Fig. 1. *Sh3Pxd2b^nee−/−^* mice are characterized by severe craniofacial dysmorphology and disrupted suture patterning.** (A) Images of C57BL/6 WT mouse and a mutant *Sh3Pxd2b^nee−/−^* mouse with craniofacial and skeletal dysmorphology with shortened nose, domed skull and smaller skeleton. (B,C) Micro-CT (B) and Alizarin Red (C) whole-mount analyses of a WT and mutant mouse calvarium at pN30 with enlarged and patent PF and Sag sutures and a shortened skull in the mutant mouse. (D,E) Alizarin Red (D) and fluorescent calcein whole-mount (E) staining of WT and *Sh3Pxd2b^nee−/−^* mouse calvarium showing a representative area of the midline of the skull from the lambdoid suture to Bregma at pN8, pN12 and postnatal months 1 and 6. Mutant mice show lack of mineralization (D) and calcification (E) in the membranous area proximal to the PF and Sag suture compared to WT mice. Ectopic bone formation is observed in the Sag suture of mutant mice at postnatal month 6 (arrow). (F) Movat's Pentachrome staining of 10 μm sections of WT and mutant PF suture pN3, 8, 12 and 17 and postnatal month 1, 2 and 6. The endocranial layer of the WT PF suture undergoes closure via endochondral ossification (arrowheads) between pN8 and pN12, whereas WT ectocranial layer of the PF suture remains patent 6 months postnatally. Dashed lines mark bone fronts. Endochondral ossification is absent in *Sh3Pxd2b^nee−/−^* mice. Images represent tiled composites of multiple adjacent fields. (G-J) Immunofluorescence staining for cartilaginous proteins characteristically present in endochondral ossification processes in physiological PF suture closure in WT mice and *Sh3Pxd2b^nee−/−^* mice. Immunofluorescent signal for SOX9 (G), COL2A1 (H), COL10A1 (I) and BGLAP (J) is present in WT but not in mutant PF sutures at pN8 and pN12. Negative controls without primary antibody are shown in the right-hand panels. (K) Movat's Pentachrome staining of 10 μm sections of WT and mutant Sag suture at pN3, 8, 12 and 17 and postnatal months 1, 2 and 6. Sag suture in WT mice remains patent through postnatal month 6. The corresponding area of the mutant Sag suture shows only suture mesenchyme with hypo-mineralization starting around pN30 (red dashed lines) and ectopic bone formation (arrowhead) at postnatal month 6. See also Fig. S1. Images represent tiled composites of multiple adjacent fields. Experiments were performed in triplicate with *n*=3 per group. A, anterior; Cor, coronal suture; f, frontal bone; JL, jugum limitans; Lam, lamboid suture; p, parietal bone; P, posterior; PF, posterofrontal suture; PFecto, ectocranial layer; PFendo, endocranial layer; Sag, sagittal suture.

calvarium at postnatal day (pN) 30 showed shortened nasal bones and enlarged, widely patent Sag and PF sutures compared to WT mice. Alizarin Red whole-mount staining was performed on mutant and WT calvariae to assess mineralization (Fig. 1C). Corresponding to micro-CT images, mutant skulls were generally shorter, and PF and Sag sutures were patent. Non-mineralized, membranous areas were present along the midline of the skull extending along the entirety of the Sag suture and in the center of the PF suture (Fig. 1B,C). To analyze mineralization and calcification of the membranous area over time, Alizarin Red (Fig. 1D) and calcein (Fig. 1E) whole-mount staining of mutant and WT calvariae were performed. No calcification or mineralization was observed in the membranous areas along the midline of the skull at any time point. At pN8, these areas extended along the entire length of the PF and Sag suture, continuously shrinking over time perpendicular to the midline (Fig. 1D,E). Interestingly, at 6 months postnatally, the Alizarin Red- and calcein-stained calvariae of the mutant mice showed ectopic bone formation (Fig. 1D, arrow) in the center of the Sag suture.

Calvarial suture development was analyzed by Movat's Pentachrome staining on serial sections to assess bone and cartilage formation. In WT PF sutures, an ectocranial and endocranial bone layer was clearly distinguishable. While the ectocranial layer remained patent, a bony bridge formed between the endocranial osteogenic fronts over time. In contrast, in $Sh3Pxd2b^{nee-/-}$ mice, the normal architecture of the PF suture was lost, endocranial and ectocranial layers were not distinguishable and the suture remained patent (Fig. 1F). As previously described (Sahar et al., 2005; Behr et al., 2010a), in WT mice the PF suture fused via endochondral ossification between pN7 and 17 (Fig. 1F, left). Conversely, neither cartilage formation nor closure was seen in the PF suture of $Sh3Pxd2b^{nee-/-}$ mice at any time (Fig. 1F, right).

The lack of endochondral ossification in mutant PF sutures was confirmed by immunofluorescence staining for specific chondrogenic markers (Fig. 1G-J). Staining with SOX9- and COL2A1-specific antibody was present in WT but not in mutant PF sutures at pN8 (Fig. 1G,H). Moreover, immunofluorescence staining for COL10A1 and BGLAP was seen at pN12 in WT but not in mutant mice (Fig. 1I,J).

The Sag suture in WT mice comprised one layer of two approximating bone fronts and remains patent through life (Behr et al., 2010a) (Fig. 1K). In contrast, the Sag suture in mutant mice displays a disorganized pattern different from the WT Sag (Fig. 1K, Fig. S1). Only suture mesenchyme with bone fronts not discernible were seen in the mutant Sag suture in the anatomical region corresponding to the area analyzed in the WT Sag suture (Fig. 1K, Fig. S1). From pN3 to 17, the suture mesenchyme was membranous and did not show signs of mineralization. Starting in the first postnatal month, faint staining appeared in the suture mesenchyme of mutant mice (Fig. S1), suggesting presence of hypo-mineralized tissue. In postnatal month 6, ectopic bone formation was observed in the mutant Sag suture (Fig. S1). Corresponding to PF suture development in mutant mice, the central membranous area of the Sag suture narrowed over time perpendicular to the midline of the skull. Higher magnification images of the central area of the Sag suture (Fig. S1, right) showed more intense staining, indicating the formation of hypo-mineralized, immature bone in the suture mesenchyme over time.

Taken together, the data reveal craniofacial deformities in $Sh3Pxd2b^{nee-/-}$ mice, such as shortened noses and domed skulls, disrupted PF and Sag suture patterning, loss of endochondral ossification and closure in PF sutures, and hypo-mineralization and ectopic bone formation in Sag sutures of $Sh3Pxd2b^{nee-/-}$ mice.

## $Sh3Pxd2b^{nee-/-}$ mice display reduced osteogenic potential *in vitro* and *in vivo*

Next, we investigated differences in osteogenic potential in WT and $Sh3Pxd2b^{nee-/-}$ mice. After 3 weeks in osteoinduction culture, we observed significantly decreased bone nodules formation and matrix mineralization in mutant osteoblasts compared to WT osteoblasts as assessed by Alizarin Red staining and quantification (Fig. 2A,B).

To assess osteoclast activity during PF suture remodeling, we performed tartrate-resistant acid phosphatase (TRAP) staining on histological sections of WT and mutant mice at pN3, 8, 12 and 30 (Fig. 2C). In WT mice, TRAP-positive osteoclasts were consistently detected along the suture margins and within resorption pits of the frontal bone plates, particularly at pN8 and pN12, indicating active bone remodeling associated with physiological suture fusion. As expected, osteoclast activity was absent within the suture mesenchyme.

In mutant mice, osteoclasts were absent at pN3, but by pN8 abundant TRAP-positive cells were observed at the bone fronts. Owing to the disrupted suture architecture in $Sh3Pxd2b^{nee-/-}$ mice, the typical osteolytic lacunae seen in the WT PF suture were absent and the entire suture region consisted of suture mesenchyme lacking TRAP-positive cells. By pN12 and pN30, osteoclasts in the mutant suture had retracted to the lateral edges of the frontal bone, and activity was diminished relative to WT, suggestive of impaired and spatially restricted osteoclastic remodeling in the mutant PF suture.

To assess *in vivo* osteogenic activity, a 2-mm calvarial defect was created in the parietal bone of WT and $Sh3Pxd2b^{nee-/-}$ mice. Defects were monitored by micro-CT imaging. Only minimal healing would be expected in 2-mm defects in parietal bones of WT mice (Behr et al., 2010b; Li et al., 2013). We observed limited bone regeneration from the periphery to the center in the defect area of WT mice over time. In contrast, the defect area in mutant mice increased over time (Fig. 2D,E). TRAP staining of histological sections of the defect area at postoperative week 18 revealed the presence of TRAP-positive cells in the bone fronts of WT and mutant mice; however, no TRAP activity was detected within the defect area itself (Fig. 2F). These findings indicate that the progressive enlargement of the defect in $Sh3Pxd2b^{nee-/-}$ mice is not due to excessive osteoclast activity at the injury site, but more likely reflects deficient osteogenic repair capacity.

Taken together, these results demonstrate that $Sh3Pxd2b^{nee-/-}$ mice exhibit reduced osteogenic potential accompanied by impaired osteoclast localization and activity during suture remodeling. The progressive enlargement of calvarial defects in mutant mice, despite the absence of excessive osteoclast activity, further supports a deficiency in osteogenic repair capacity.

## Altered intracellular spatial distribution of TKS4 in neural crest-derived dura mater cells and decreased TKS4 expression in neural crest-derived PF suture mesenchyme in $Sh3Pxd2b^{nee-/-}$ mice

By performing immunofluorescence staining, we investigated TKS4 protein levels and distribution in WT and mutant dura mater cells and osteoblasts. Dura mater cells play a crucial role in embryonic skull development, in addition to contributing to suture fusion and calvarial reossification postnatally (Hobar et al., 1993; Spector et al., 2002; Bradley et al., 1997; Jiang et al., 2002). In WT dura mater cells, we observed uniform nuclear and perinuclear distribution of TKS4 (Fig. 3A, white arrows), extending in a radial pattern into the cytoplasm (Fig. 3A, yellow arrowheads). In comparison, in mutant dura mater cells nuclear staining was less

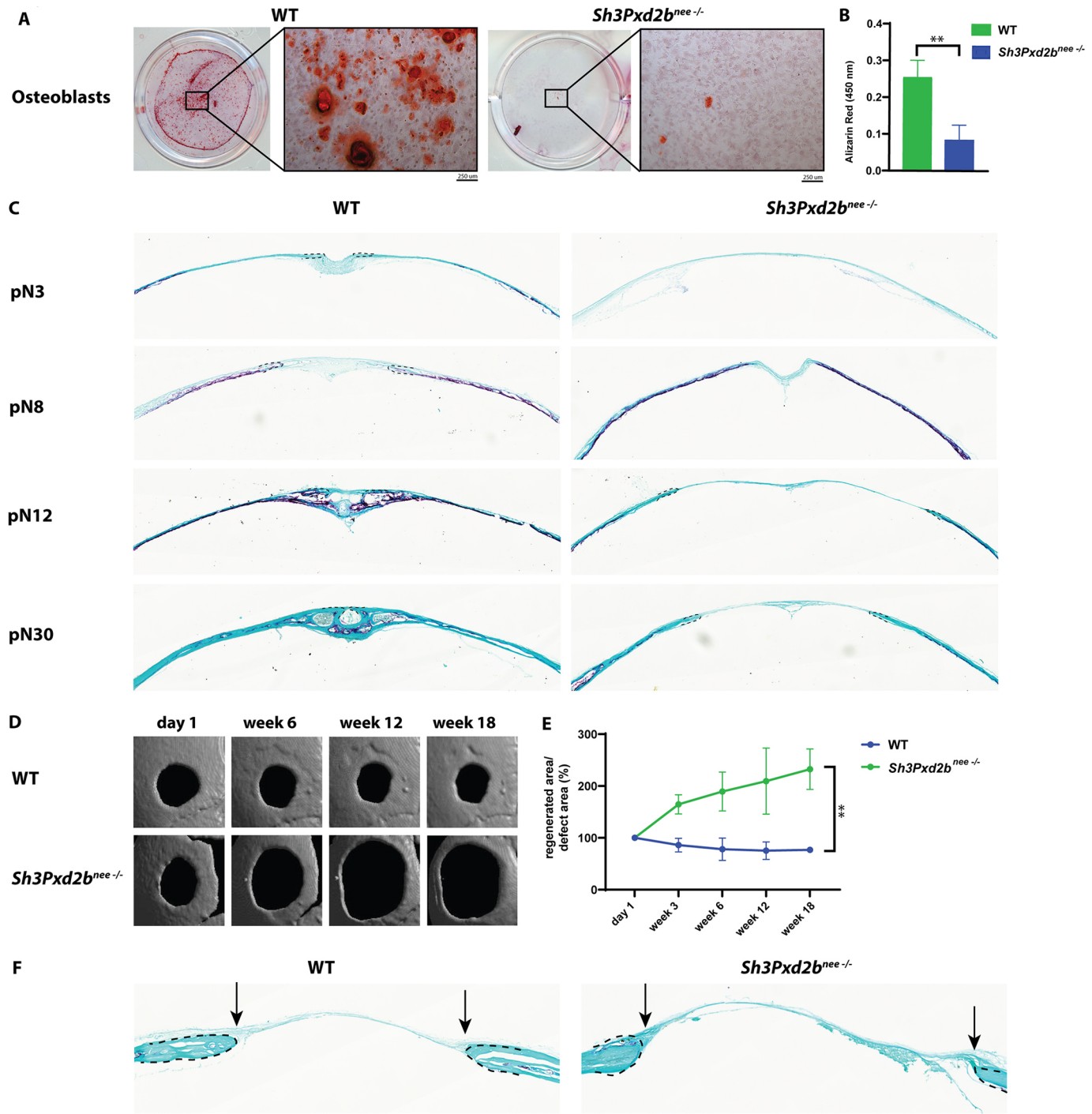

**Fig. 2. Sh3Pxd2b$^{nee-/-}$ mice display reduced osteogenic potential *in vitro* and *in vivo*.** (A) Alizarin Red staining in WT and Sh3Pxd2b$^{nee-/-}$ osteoblasts showing decreased bone nodule formation and extracellular matrix mineralization in mutant compared to WT cells undergoing osteogenic differentiation. (B) Quantification of Alizarin Red stain in WT and mutant osteoblasts. (C) TRAP staining on histological sections of WT and mutant PF suture at pN3, 8, 12 and 30 showing decreased staining in suture area of mutant mice compared to WT mice starting at pN12. Dashed lines mark bone fronts. Images represent automatically tiled composites of multiple adjacent fields acquired with a digital slide scanner. (D) Micro-CT images of 2-mm calvarial defects in WT and Sh3Pxd2b$^{nee-/-}$ mice at day 0, and week 6, 12 and 18 postoperatively showing minimal bone regeneration in WT mice and an increase in defect area in Sh3Pxd2b$^{nee-/-}$ mice over time. (E) Summary line graph showing the defect area over time as measured by percentage of decrease/increase of defect area compared to initial size of defect area. (F) TRAP staining of histological sections of the defect area at week 18 showing similar staining patterns in WT and mutant mice. Arrows denote defect area. Images represent automatically tiled composites of multiple adjacent fields acquired with a digital slide scanner. Experiments were performed in triplicate with $n=3$ per group. Data shown as mean (±s.d.). **$P<0.01$ (ANOVA, Šidák's multiple comparisons test).

intense. TKS4 appeared to accumulate in perinuclear regions, condensing in focal areas (Fig. 3A, white arrowheads) with less cytoplasmic expression. In osteoblasts, less nuclear staining was observed compared to dura mater cells. In WT osteoblasts, in a similar pattern as in dura mater cells, TKS4 was uniformly expressed perinuclear (Fig. 3B, white arrows), whereas staining in

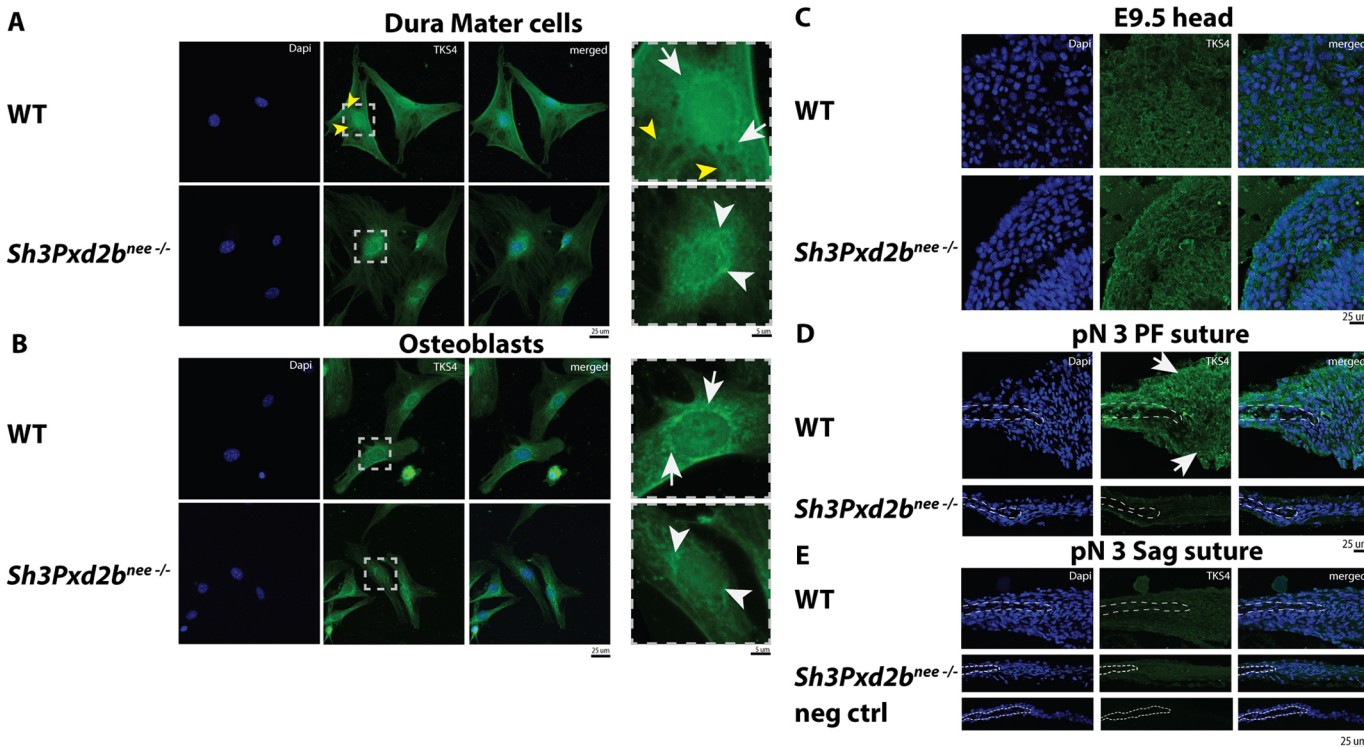

**Fig. 3. Altered intracellular spatial distribution of TKS4 in neural-crest derived dura mater cells and decreased TKS4 expression in neural crest-derived PF suture mesenchyme in _Sh3Pxd2b_[nee−/−] mice.** (A,B) Immunocytochemistry for TKS4 expression in WT and mutant dura mater cells (A) and osteoblasts (B). Boxed areas are shown as magnified representative areas of TKS4 expression in the right-hand panels. (A) Uniform nuclear and perinuclear distribution of TKS4 (white arrows) in WT dura mater cells, extending radially into the cytoplasm (yellow arrowheads) and less-intense nuclear staining and perinuclear, focally condensed accumulation in _Sh3Pxd2b_[nee−/−] dura mater cells (white arrowheads). (B) Uniform perinuclear TKS4 expression in WT osteoblasts (white arrows) compared to focally condensed perinuclear accumulation in mutant cells (white arrowheads). (C-E) Immunofluorescence staining for TKS4 in histological sections of the presumptive head area of E9.5 (C; see Fig. S2A for anatomical landmarks) and pN3 WT and mutant PF (D) and Sag sutures (E). (C) TKS4 staining is similar in WT and mutant E9.5 head areas. (D,E) TKS4 staining is observed in WT PF suture mesenchyme, periosteum and dura mater (white arrows) and in Sag sutures, but not in mutant sutures. Negative control for TKS4 without primary antibody is shown in the bottom panels of E. Bone fronts are marked by white dashed lines. Experiments were performed in triplicate with _n_=3 per group.

mutant osteoblasts appeared more condensed focally (Fig. 3B, white arrowheads). In over 35 counted mutant cells, perinuclear condensation of TKS4 staining was observed in 75% and 57% of dura mater cells and osteoblasts, respectively. Taken together, these findings indicate altered intracellular TSK4 distribution in _Sh3Pxd2b_[nee−/−] mice, which is more pronounced in dura mater cells.

TKS4 immunofluorescence staining on histological sections of the presumptive area of the head in embryonic day (E) 9.5 mutant and WT embryos did not show any differences in TKS4 levels (Fig. 3C, Fig. S2A), while decreased TKS4 staining was observed in the suture mesenchyme, periosteum and dura mater of mutant PF suture compared to WT mice (Fig. 3D, white arrows). Given the severe disruption of PF suture architecture in the mutants, the reduction in TKS4 signal may reflect the partial absence or disorganization of specific cell populations. No difference of TKS4 staining was seen between WT and mutant Sag sutures (Fig. 3E).

### _Sh3Pxd2b_[nee−/−] mice are characterized by decreased proliferation and migration _in vitro_ and _in vivo_

To understand the potential mechanisms governing the craniofacial phenotype of _Sh3Pxd2b_[nee−/−] mice, we assessed proliferation, apoptosis and migration in WT and mutant mice _in vitro_ and _in vivo_.

The 5-ethynyl-2′-deoxyuridine (EdU) proliferation assay revealed significantly decreased proliferative activity in mutant compared to WT cells with 40% versus 9% and 37% versus 16% of EdU-positive

osteoblasts and dura mater cells, respectively ($P<0.05$; Fig. 4A-C). To assess proliferation _in vivo_, immunofluorescence staining for PCNA was performed in tissue sections of the presumptive head area of E9.5 embryos and pN3 PF and Sag sutures. In E9.5 WT embryos, PCNA-positive cells were prominently observed in the surface ectoderm and adjacent mesenchymal layers of the presumptive head, while these regions showed markedly reduced staining in mutant embryos (Figs S2A, Fig. 4D, arrow). In WT PF and Sag sutures, proliferation was generally observed in the bone plates (Fig. 4E, dashed lines) and suture mesenchyme, including the periosteum and dura mater (Fig. 4E, arrows). In mutant PF and Sag sutures, similar PCNA staining was observed in the bone plates, whereas no staining was detected within the suture mesenchyme (Fig. 4E,F).

Conversely, no differences were observed in apoptotic activity assessed by terminal deoxynucleotidyl transferase-mediated dUTP nick-end labeling (TUNEL) assay in dura mater cells and osteoblasts _in vitro_ and _in vivo_ (Fig. S2B-F).

TKS4 is required for podosome formation in migratory cells (Buschman et al., 2009). To establish whether the craniofacial phenotype of _Sh3Pxd2b_[nee−/−] mice is characterized by cell migratory defects, a scratch assay was performed in WT and mutant dura mater and osteoblast cell cultures. Images of cell migration showed decreased migration in mutant dura mater cells at 24 h compared to WT cells (Fig. 4G, left) and similar migration patterns in WT and mutant osteoblasts (Fig. 4G, right). Cumulative data of three assays from 0 h to 62 h are depicted as a line graph in Fig. 4H.

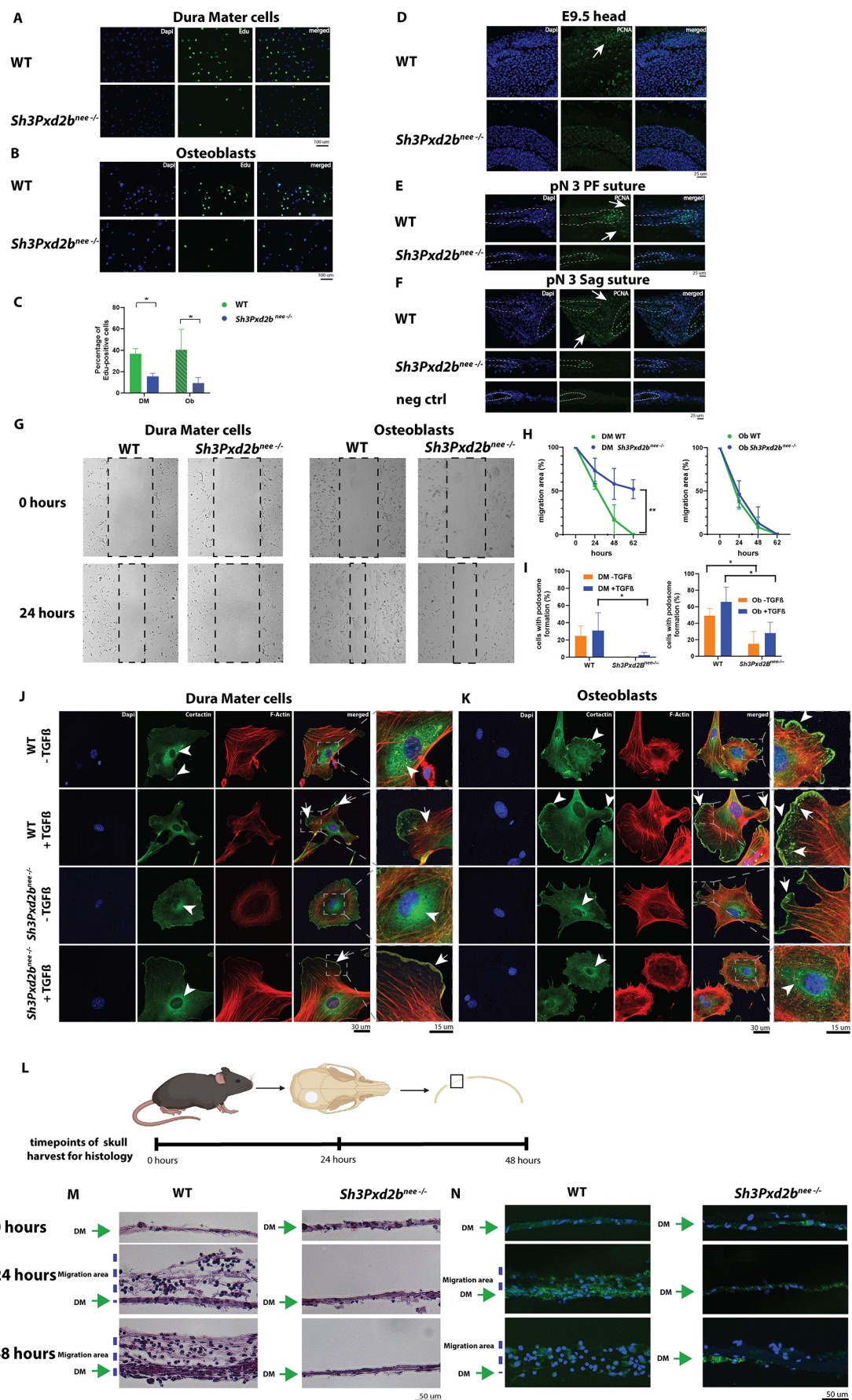

**Fig. 4.** See next page for legend.

**Fig. 4. Sh3Pxd2b**$^{nee-/-}$ **mice are characterized by decreased proliferation and migration *in vitro* and *in vivo*.** (A,B) EdU proliferation assay shows decreased proliferation in mutant (bottom) dura mater cells (A) and osteoblasts (B) compared to WT cells (top). (C) Percentage of EdU-positive WT and mutant osteoblasts and dura mater cells. (D-F) Immunofluorescence staining for PCNA in the presumptive head area of E9.5 (D; see Fig. S2A for anatomical landmarks) and pN3 WT and mutant PF (E) and Sag (F) sutures. (D) Decreased proliferative activity in surface ectoderm and adjacent mesenchymal layers of mutant compared to WT E9.5 head area (arrows). (E,F) Decreased PCNA staining in mutant compared to WT PF and Sag suture mesenchyme, periosteum and dura mater and similar activity in bone fronts (dashed lines). Negative control for PCNA without primary antibody is shown in the bottom panels of F. (G) Scratch assay to assess *in vitro* migration of dura mater cells and osteoblasts from WT and *Sh3Pxd2b*$^{nee-/-}$ mice at 0 and 24 h, demonstrating decreased migration in mutant dura mater cells. Dashed boxes indicate scratch area reducing over time. (H) Summary line graphs quantifying cell migration as percentage of decrease of initial scratch area. Significantly decreased migration is observed in mutant compared to WT dura mater (DM) cells. Migration in WT and mutant osteoblasts (Ob) is similar. (I) Quantification of dura mater cells and osteoblasts with podosome formation with/without TGFβ as a percentage of all counted cells showing significantly decreased number of dura mater cells with podosome formation mutant compared to WT mice, upon TGFβ stimulation (left graph), and significantly decreased number of osteoblasts with podosome formation in stimulated and unstimulated mutant compared to WT cells (right graph). (J,K) Immunocytochemistry for cortactin, F-actin and DAPI in dura mater cells and osteoblasts with/without TGFβ from WT and *Sh3Pxd2b*$^{nee-/-}$ mice. Boxed areas showing regions of interest are shown at higher magnification on the right. (J) Increased perinuclear accumulation is observed in unstimulated WT cells (arrowheads, magnified panel). Upon TGFβ stimulation in WT cells, cortactin spreads out in the cytoplasm. Actin-rich puncta colocalize with cortactin as indicated by yellow staining, localized in small clusters near the cell membranes and in cell protrusions of stimulated WT cells (arrows, magnified panel). In mutant unstimulated cells, cortactin accumulates in perinuclear regions (arrowheads, magnified panel). Cell membranes appear frayed. Upon TGFβ stimulation, cortactin colocalizes with actin at the cell membranes (arrows, magnified panel). (K) In WT osteoblasts, intense cortactin staining is observed along the cell membranes (arrowheads, magnified panel). Upon TGFβ stimulation, cortactin spreads out in the cytoplasm towards large cell protrusions with multiple actin-rich puncta colocalizing with cortactin (arrows, magnified panel). In unstimulated and stimulated mutant osteoblasts, cortactin accumulates in perinuclear regions (arrowheads). Small puncta form unevenly distributed in the cytoplasm but do not colocalize with actin and do not accumulate near the cell membrane (arrow). Cell protrusions do not form. (L) Schematic depicting creation of calvarial defects in mouse parietal bones and harvesting of skulls at 0, 24 and 48 h postoperatively. Created in BioRender by Huber, J., 2026. https://BioRender.com/h47f781. This figure was sublicensed under CC BY 4.0 terms. (M) Hematoxylin and Eosin-stained 10 μm sections of representative calvarial defect area in WT and *Sh3Pxd2b*$^{nee-/-}$ mice at 0, 24 and 48 h postoperatively. At 24 and 48 h, cells have migrated into the defect area in WT mice, whereas in mutant mice no migration of cells can be observed. See also Fig. S3. (N) 10 μm sections of representative calvarial defect area in *Wnt1-Cre2*$^{+/-}$;*mT/mG* and *Wnt1-Cre2*$^{+/-}$;*mT/mG;Sh3Pxd2b*$^{nee-/-}$ mice showing migration of Wnt1-Cre2-positive cells expressing green fluorescent protein into the defect area in WT mice and no migration in mutant calvarial defects, respectively. In M and N, dura mater is marked with green arrows and migration area with blue dashed lines. Experiments were performed in triplicate with *n*=3 per group, when applicable. Data shown as mean (±s.d.). *P<0.05; **P<0.01 (ANOVA, Šidák's multiple comparisons).

Next, to investigate whether dura mater cells and osteoblasts would form podosomes, we performed immunofluorescence staining for the podosomal markers cortactin and F-actin in non-migratory cells and cells induced with TGFβ, a stimulator of cell migration (Murphy et al., 2011). The percentage of cells with podosome formation was significantly decreased in TGFβ-stimulated mutant dura mater cells compared to TGFβ-stimulated WT dura mater cells (Fig. 4I, left graph). The percentage of

osteoblasts with podosome formation was significantly decreased in mutant compared to WT mice (stimulated and not stimulated; Fig. 4I, right graph) (*P*<0.05). In unstimulated WT dura mater cells, cortactin accumulated uniformly in perinuclear regions and near the cell membranes (Fig. 4J, arrowheads). Upon TGFβ stimulation, cortactin extended radially through the cytoplasm toward the leading edge, forming actin-rich puncta at the cell protrusions (Fig. 4J arrows, yellow staining). Unstimulated mutant dura mater cells showed uneven cortactin accumulation in perinuclear regions, with pronounced focal staining (Fig. 4J, arrowhead). Cell protrusions were absent, and cell membranes appeared frayed. After TGFβ stimulation, mutant dura mater cells formed protrusions, and cortactin and actin colocalized along the cell membrane (Fig. 4J, arrows), while also showing enhanced perinuclear accumulation (Fig. 4J, arrowhead) and no cytoplasmic distribution (Fig. 4J).

In WT osteoblast cells, we observed intense cortactin staining along the cell membranes (Fig. 4K, arrowheads). Upon TGFβ stimulation, cortactin extended into the cytoplasm and large cell protrusions formed, marked by multiple actin-rich puncta at the cell edges colocalizing with cortactin (Fig. 4K, arrows, yellow staining). In contrast, unstimulated and stimulated mutant osteoblasts displayed perinuclear cortactin accumulation (Fig. 4K, arrowheads), with small, unevenly distributed cortactin-rich puncta in the cytoplasm that did not colocalize with F-actin or localize to the cell membrane (Fig. 4K, arrow). Cell protrusions did not form (Fig. 4K).

To assess cell migration *in vivo*, we created a 2-mm circular defect in the parietal bone of WT and *Sh3Pxd2b*$^{nee-/-}$ mice and harvested skulls at 0, 24 and 48 h postoperatively (Fig. 4L). In WT mice, we observed migration of cells into the defect area at 24 and 48 h (Fig. S3, Fig. 4M, left). In *Sh3Pxd2b*$^{nee-/-}$ mice, cells did not migrate into the defect area (Fig. S3, Fig. 4M, right). The *Wnt1-Cre2* transgenic mouse line is used to express Cre recombinase in neural crest cell lineages (Dinsmore et al., 2022). We repeated the experiment in *Wnt1-Cre2*$^{+/-}$;*mT/mG* and *Wnt1-Cre2*$^{+/-}$;*mT/mG; Sh3Pxd2b*$^{nee-/-}$ mice, confirming that the cells migrating into the defect area in WT mice expressed green fluorescent protein and were thus of neural crest origin. Conversely, migration into the defect area did not occur in *Wnt1-Cre2*$^{+/-}$;*mT/mG;Sh3Pxd2b*$^{nee-/-}$ mice (Fig. 4N).

The above data reveal decreased and spatially altered proliferative and decreased migratory activity in *Sh3Pxd2b*$^{nee-/-}$ mice, which is more pronounced in cells and tissues of neural crest origin. Additionally, mutant dura mater cells and osteoblasts form fewer and structurally aberrant podosomes necessary for cell migration, compared to WT cells, thus confirming the findings of previous studies establishing the role of TKS4 in podosome formation (Bögel et al., 2012; Buschman et al., 2009; Courtneidge, 2012; Iqbal et al., 2010; Kudlik et al., 2020; Massadeh et al., 2022).

## Migration and podosome formation in neural crest-derived cells and tissues from *Sh3Pxd2b*$^{nee-/-}$ mice is impaired

To gain insights into the role of TKS4 in cranial neural crest cells (NCCs), we isolated the neural plate (NP) from E9.5 mouse embryos to assess TKS4 expression and migratory potential. The presence of migratory NCCs was confirmed by immunofluorescence staining for SOX10 (Trainor, 2010) (Fig. 5A, Fig. S2A). TKS4 expression was unaffected in the NP of WT and *Sh3Pxd2b*$^{nee-/-}$ mice *in vivo* (Fig. 5B, Fig. S2A). Proliferation was generally decreased in the NP of mutant mice compared to WT mice. Interestingly, proliferation was more enhanced in the outer layer and focally present in the sub-layers of the NP of mutant mice, possibly indicating the presence of subpopulations of proliferative cells specific to the *Sh3Pxd2b*$^{nee-/-}$

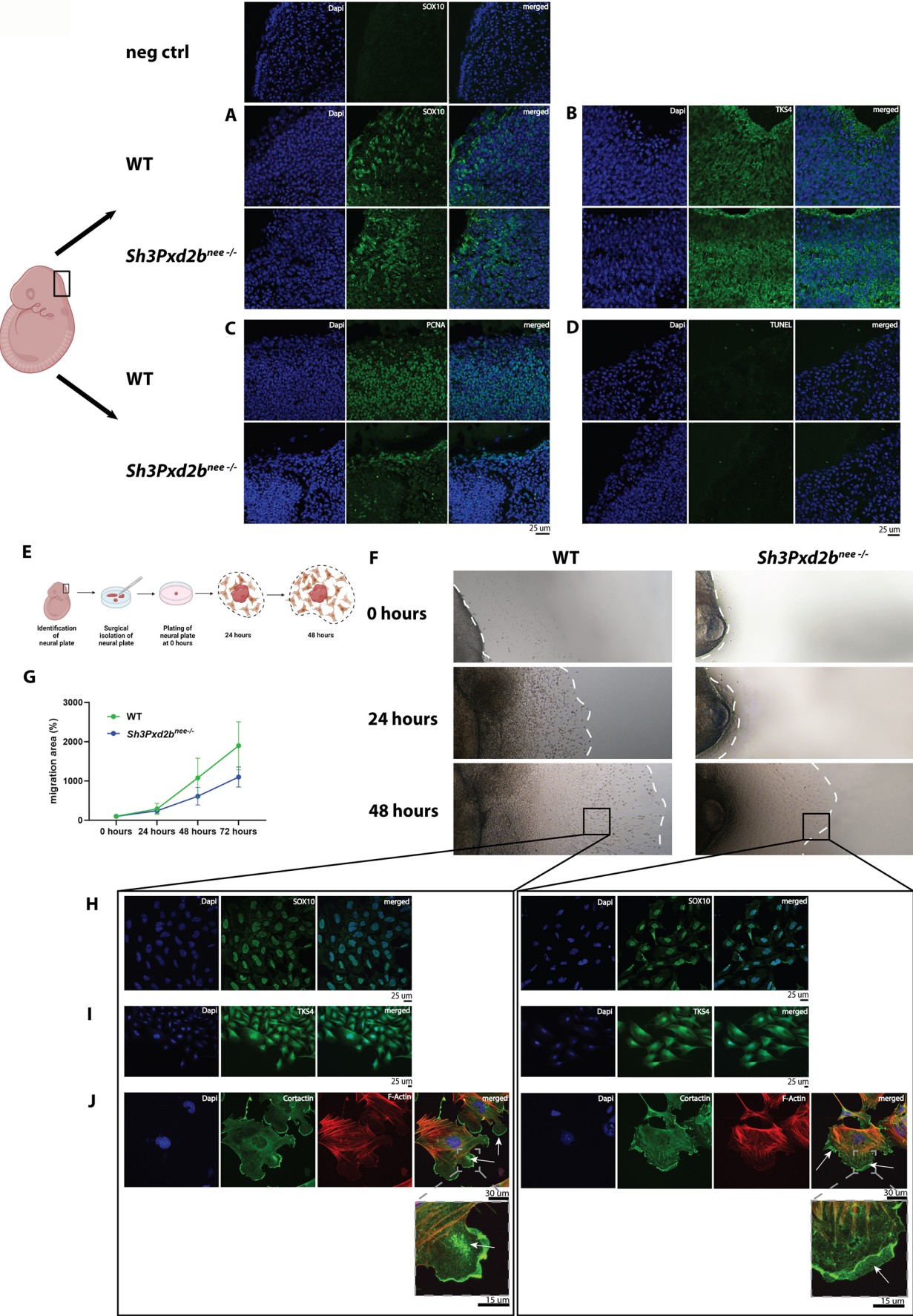

**Fig. 5.** See next page for legend.

**Fig. 5. Migration and podosome formation in neural crest-derived cells and neural crest tissues from *Sh3Pxd2b^nee−/−* mice is impaired.** (A-D) Illustration of a mouse embryo at E9.5 with a rectangle indicating the area of the neural plate (NP). See Fig. S2A for additional anatomical landmarks. Images show immunofluorescence staining of tissue sections of the NP for SOX10 (A), TKS4 (B), PCNA (C) and TUNEL assay (D). Staining of SOX10 confirms the presence of neural crest cells (NCCs). Top panels depict negative control for SOX10 without primary antibody. TKS4 expression is comparable in WT and mutant tissues. PCNA staining is more enhanced in the outer layers and focally present in the sub-layers of the mutant NP. TUNEL assay shows no difference of apoptotic activity between WT and mutant NP. (E) Schematic depicting the dissection and plating of the NP and migration of NCCs. Created in BioRender by Huber, J., 2026. https://BioRender.com/h95h065. This figure was sublicensed under CC BY 4.0 terms. (F) Migrating NCCs observed at 0 h and after 24, 48 and 72 h in culture. White dashed lines mark the end of the migrated area. Migration is decreased in mutant NCCs. (G) Histogram demonstrating increase of the migrated area over time as percentage increase relative to the size of the plated tissue. Data shown as mean (±s.d.). (H-J) Representative areas of migrating cells were stained for SOX10 (H) to confirm neural crest origin and TKS4 (I) showing comparable expression and distribution in WT and mutant cells. Podosome formation assay with cortactin and F-actin was performed (J) showing formation of protrusions in WT cells with accumulation of cortactin in clusters and lining the cell membrane (arrows). Mutant cells show cortactin near cell membranes, but cell boundaries appear frayed and podosomes are not formed. Experiments were performed in triplicate.

mouse (Fig. 5C, Fig. S2A). No differences in apoptotic activity were observed in the NP of WT and mutant mice (Fig. 5D, Fig. S2A).

Next, the NP from E9.5 embryos was isolated and plated as previously described (Gonzalez Malagon et al., 2019) (Fig. 5E). The NP and the peripherally migrating cells were imaged at time of plating (0 h), at 24, 48 and 72 h and stained for SOX10, TKS4 and cortactin. Interestingly, migration of mutant NCCs over 72 h was slower than migration of WT NCCs (Fig. 5F,G). Nuclear expression of SOX10 in cells isolated from WT and mutant migrating cells confirmed neural crest origin (Fig. 5H). TKS4 was similarly expressed in NCCs from WT and *Sh3Pxd2b^nee−/−* mice (Fig. 5I). Analyzing podosome formation of migrating NCCs, we showed formation of protrusions and accumulation of cortactin cluster in the protrusions and along the cell membranes of WT NCCs (Fig. 5J, white arrows, magnified panel). In contrast, in mutant NCC, increased cortactin staining was seen near the cell membranes, which appeared frayed. The cytoskeleton was in disarray and the cell membranes seemed to collapse, unable to form a structured protrusion (Fig. 5J, white arrows, magnified panel).

## Downregulation of ribosomal genes expression and accumulation of ribosomal RNA in cell protrusions of *Sh3Pxd2b^nee−/−* mice

Data gained from our investigation indicate that the craniofacial phenotype of *Sh3Pxd2b^nee−/−* mice associates with decreased cell proliferation and migration. Although impaired podosome formation provides a valid explanation for the migratory deficiency of cells, it does not explain the lack of proliferative activity, nor does it explain the tissue specificity of the craniofacial phenotype observed in *Sh3Pxd2b^nee−/−* mice.

To gain more insights into the role of *Sh3Pxd2b*, we performed bulk RNA sequencing transcriptomic analysis on whole-mount skulls from pN15 mice. In principal component analysis, samples from WT and mutant mice clustered distinctly, with more dispersion in the WT group (Fig. 6A). The volcano plot in Fig. 6B demonstrates a greater number of significantly upregulated than downregulated genes in the mutant group compared to the WT group. An adjusted *P*-value cut-off of 0.05 and a FC cut-off of ±1 was chosen.

Gene set enrichment analysis revealed enrichment of ribosome and ribosome biogenesis pathways among all genes downregulated in mutant mice (Fig. 6C, Table S1). Among the top 50 downregulated genes in mutant skulls were C/D box small nucleolar RNA (SNORD) genes and nuclear-encoded ribosomal RNA (rRNA) 5S, a component of the large ribosome 60S subunit (Fig. 6B,D).

Agilent Bioanalyzer analysis of RNA extracted from skull tissue revealed a generally lower concentration of small RNAs (757.6±411.1 pg/μl versus 3293.1±2106.1 pg/μl; mean±s.d.) and reduced proportion of microRNA (5.7±2.1% versus 18.2±6.8%) in *Sh3Pxd2b^nee−/−* mice compared to WT (Fig. 6E, Table S2).

A detailed network analysis of downregulated genes involved in ribosome biogenesis is illustrated in Fig. 6F. Genes are grouped according to their molecular function. The main gene hubs include genes involved in ribosomal large (e.g. RPL genes) and small (e.g. RPS genes) subunit biogenesis, ribosome assembly (e.g. *Eif2a*) and ribosome biogenesis.

The top 50 ranked upregulated genes (Table S3) are involved in mostly metabolic and biochemical pathways, apparently unrelated to craniofacial development, and therefore were not further analyzed for the purposes of this study.

Next, to investigate the intracellular localization of rRNAs and possible colocalization with TKS4, we performed immunofluorescence staining for TKS4 and Y10b, a monoclonal antibody detecting rRNA in dura mater cells, osteoblasts and NCCs from WT and *Sh3Pxd2b^nee−/−* mice (Fig. 7A-C) and in the presumptive head region and NP of E9.5 embryos (Fig. 7E,F). In WT dura mater cells, we observed a homogeneous cytoplasmic distribution of Y10b and focal presence in membrane protrusions and typical extension structures of the plasma membrane, such as circular dorsal ruffles (Fig. 7A, magnified panels). This suggests that structures of the plasma membrane are involved in ribosomal translational processes, which is in accordance with other studies (Gabanella et al., 2020). In contrast, in dura mater cells from *Sh3Pxd2b^nee−/−* mice, Y10b and TKS4 accumulated perinuclear (Fig. 7A, magnified panels) and cytoplasmic distribution was limited. In WT osteoblasts, TKS4 was similarly present in the cytoplasm. Y10b showed a radial distribution from perinuclear extending towards the cell membrane (Fig. 7B, magnified panels). In mutant osteoblasts, TKS4 and Y10b also accumulated in perinuclear regions. In WT NCCs, strong immunoreactivity for TKS4 and Y10b was seen in the cytoplasm, extending towards a membrane protrusion (Fig. 7C, magnified panels). Interestingly, in NCCs from *Sh3Pxd2b^nee−/−* mice, we observed no perinuclear accumulation of Y10b as in osteoblasts and dura mater cells, but rather intense focal reactivity at the cell membrane (Fig. 7C, arrows, magnified panels).

The Pearson's coefficient quantifies the degree of linear correlation between the intensity values of two fluorophores. Here, we observed a strong positive correlation between the two fluorophores of TKS4 and Y10b with a similar coefficient among all cell types with a median of 0.78 (range 0.02, 0.12), suggesting substantial overlap of TKS4 and Y10b (Fig. 7D) (Bolte and Cordelières, 2006).

Immunofluorescence staining for Y10b in the presumptive head area and NP of E9.5 embryos (Fig. 7E,F, Fig. S2A) showed uniform staining in WT and mutant embryonic heads with slightly enhanced expression of Y10b in the mutant mouse (Fig. 7E). Compared to WT, Y10b staining was decreased in the inner layers of the NP (Fig. 7F), mirroring the PCNA staining shown in Fig. 4D.

Taken together, our sequencing results show that *Sh3Pxd2b^nee−/−* mice exhibit downregulation of multiple genes involved in ribosome

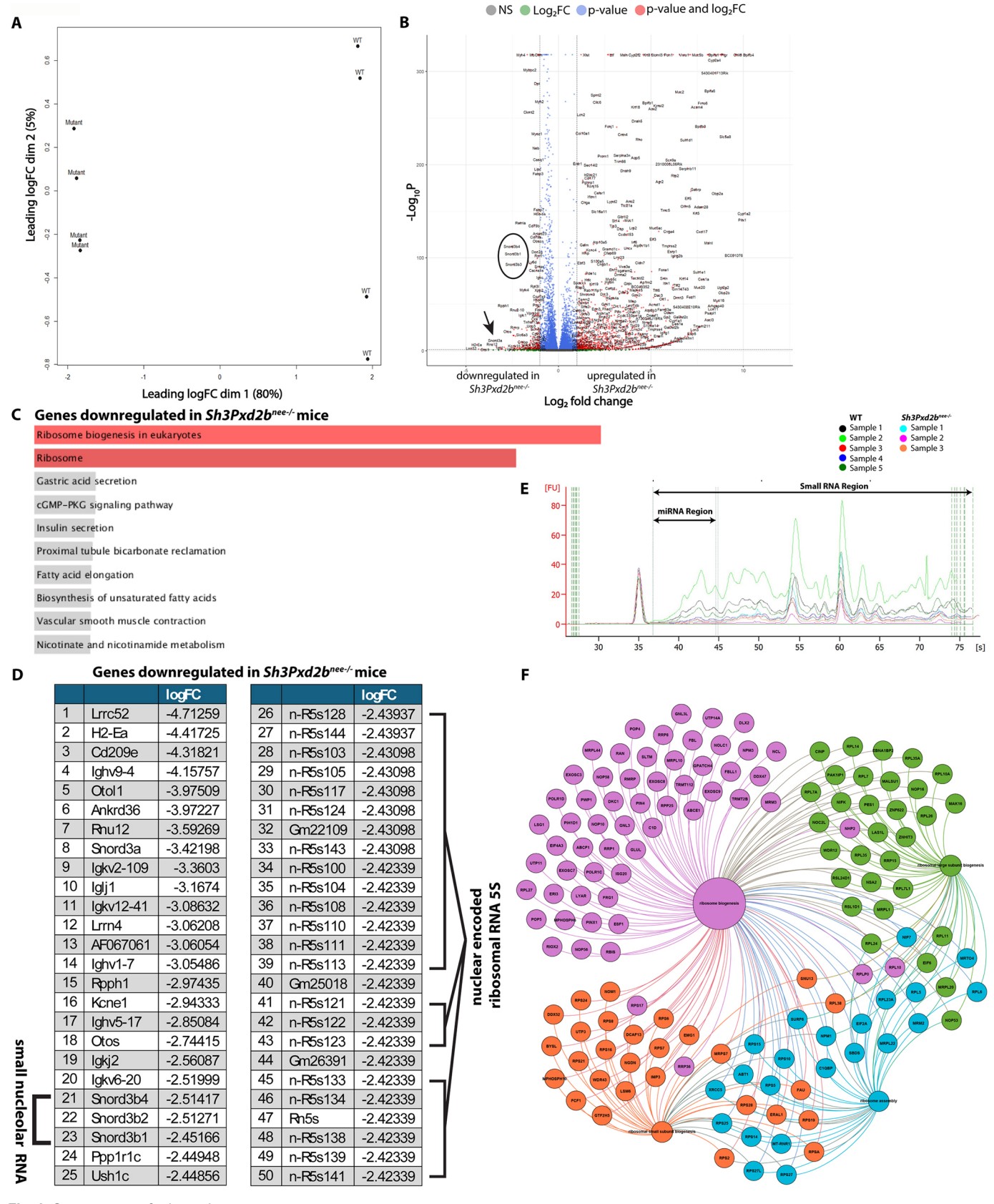

**Fig. 6.** See next page for legend.

biogenesis. Bioanalyzer analysis showed reduced concentration of small RNA and microRNA levels in *Sh3Pxd2b*$^{nee-/-}$ mice. Immunofluorescence revealed altered subcellular localization of rRNA, with Y10b signal showing focal enrichment at membrane protrusions in NCCs of mutant mice. TKS4 showed substantial spatial overlap with Y10b across all examined cell types.

**Fig. 6. Ribosome biogenesis is impaired.** (A) Principal component analysis with 80% variance in PC1 and 5% variance in PC2, demonstrating clear differences between the WT and mutant mice samples. (B) Volcano plot showing log$_2$(FC) versus −log$_{10}$(P-value) with an FC cut-off of ±1 and P-value cut-off of 0.05. We observed greater number of significantly upregulated than downregulated genes in the mutant group compared to the WT group. Circle and arrow mark genes of interest. (C) Gene set enrichment analysis using KEGG pathway database 2019 of differentially expressed genes between WT and *Sh3Pxd2b*$^{nee−/−}$ mice showing enrichment of ribosome and ribosome biogenesis pathways. (D) Table depicting the 50 most differentially expressed genes downregulated in the mutant mouse gene sets, revealing downregulation of SNORD genes and nuclear-encoded ribosomal RNA 5S (e.g. n-R5s128) in skulls of *Sh3Pxd2b*$^{nee−/−}$ mice. n=3 per group in RNA bulk sequencing. (E) Overlaid electropherograms from Agilent Bioanalyzer small RNA assay showing reduced small RNA concentration and microRNA proportion in *Sh3Pxd2b*$^{nee−/−}$ skulls at pN15. (F) Network analysis of downregulated genes involved in ribosome biogenesis grouped by molecular function.

## DISCUSSION

FTHS has a poorly characterized pathology and the underlying biomolecular mechanism(s) are largely unknown. Findings stemming from our study shed light on this syndrome and establish the *Sh3Pxd2b*$^{nee−/−}$ mouse as an attractive animal model to unveil new aspects of this disease.

In this mouse model, a frame-shift mutation in the *Sh3Pxd2b* gene causes a truncation of the encoded protein TKS4. In this study, we observed aberrant intracellular TKS4 expression in mutant dura mater cells and osteoblasts, including increased perinuclear condensation and altered cytoplasmic expression. While the biological significance of this distribution pattern remains to be fully determined, it is notable that previous studies have demonstrated a similarly altered intracellular distribution pattern and accumulation of mutated TKS4 in aggresomes, which are typically located perinuclear and play a key role in eliminating misfolded proteins (Ádám et al., 2015). These findings suggest that in the *Sh3Pxd2b*$^{nee−/−}$ mouse, spatial localization of the mutant TKS4 protein may indicate misfolding and thus functional impairment contributing to the observed craniofacial phenotype.

The first part of our study provides a detailed description of previously undocumented features of the craniofacial phenotype of the *Sh3Pxd2b*$^{nee−/−}$ mouse. Several craniofacial dysmorphologies are identified, among them widened fontanels, severe impairment of Sag and PF suture patterning and anatomical architecture. Additionally, the osteo-regenerative potential is greatly reduced in *Sh3Pxd2b*$^{nee−/−}$ mice.

The second part of the study unveils potential biomolecular mechanisms paralleling the craniofacial dysmorphic phenotype of *Sh3Pxd2b*$^{nee−/−}$ mice. Data gained from our investigation demonstrate a decreased and spatially altered osteoblast and dura mater cell proliferation concomitant to dysfunctional podosome formation and impaired cell migration both *in vitro* and *in vivo*. Similar findings are also found in the NP and cranial NCCs. Thus, our data are suggestive of abnormalities in the cephalic neural crest lineage in *Sh3Pxd2b*$^{nee−/−}$ mice. Of note, Murphy et al. (2011) have demonstrated that the zebrafish homolog Tks5 is required for neural crest cell podosome formation and for neural crest cell migration during embryogenesis, thus supporting our finding for TKS4. This impaired migration likely contributes to the observed reduction in local proliferation, potentially due to impaired recruitment or maintenance of specific cell populations.

In light of these observations, the impaired migration of the cephalic neural crest cell lineage in *Sh3Pxd2b*$^{nee−/−}$ mice would explain the patterning disruption and the lack of endochondral

ossification leading to absence of closure of the PF suture, which is of neural crest origin (Jiang et al., 2002).

Moreover, we observed decreased osteogenesis *in vitro* and decreased osteoregeneration in calvarial defects *in vivo* in *Sh3Pxd2b*$^{nee−/−}$ mice. Over time, calvarial defects increased in size compared to defects in WT mice, suggesting an imbalance in bone homeostasis in *Sh3Pxd2b*$^{nee−/−}$ mice. TRAP staining indicated that this was most likely due to deficient osteogenic repair capacity rather than excessive osteoclast activity. Previous studies have demonstrated that TKS4 plays an important role in mesenchymal stem cell differentiation into the osteoblast lineage, while number and function of osteoclasts remained intact, ultimately leading to a cellular imbalance in favor of osteoclasts underlying the bone phenotype of *Sh3Pxd2b*$^{nee−/−}$ mice (Vas et al., 2019; Dülk et al., 2016). However, osteoclasts are highly migratory cells and rely on podosome formation to migrate to the area of bone resorption and establish sealing zones maintaining a highly resorptive microenvironment (Georgess et al., 2014). In the context of *Sh3Pxd2b*$^{nee−/−}$ mice, TKS4 may play a dual role by supporting podosome-mediated migration and function of osteoclasts, as well as promoting differentiation of mesenchymal stem cells into osteoblasts. Loss of TKS4 function could therefore disrupt one or both arms of bone homeostasis, leading to defective bone regeneration. TRAP staining suggests that osteoclast activity is impaired or spatially restricted in mutant mice. Further studies are needed to determine whether this reflects intrinsic dysfunction of osteoclasts or impaired recruitment due to defective migration.

In humans, besides FTHS, other genetic diseases have been associated with de-regulation of podosomal components (Linder et al., 1999; Calle et al., 2008; Cejudo-Martin and Courtneidge, 2011; Welsch et al., 2009; Ayala et al., 2009). The similarities in craniofacial deformities of these diseases imply a common underlying pathophysiology suggestive of podosome malfunction (Cejudo-Martin and Courtneidge, 2011; Saini and Courtneidge, 2018). Diseases associated with impaired neural crest cell migration in embryonic development also share similar craniofacial features (Acloque et al., 2009; Cejudo-Martin and Courtneidge, 2011; Wu et al., 2017).

However, the decreased cell proliferation in the *Sh3Pxd2b*$^{nee−/−}$ mouse is suggestive of additional causal processes contributing to the craniofacial phenotype.

Multiple studies have shown that ribosomal protein-mRNAs (RP-mRNAs) and rRNA localize to actin-rich cell protrusions of migrating cells, leading to an increase in local translation and upregulation of ribosome biogenesis (Dermit et al., 2020; Mili et al., 2008; Mardakheh et al., 2015; Gabanella et al., 2020). This process is crucial to ensure sustained growth and migration and is universal to all migratory cells (Dermit et al., 2020).

Our transcriptomic analysis revealed downregulation of multiple genes involved in ribosome biogenesis, including SNORD genes and nuclear-encoded rRNA 5S. Agilent Bioanalyzer analysis revealed a reduction in both small RNA concentration and relative microRNA proportion. While these findings do not directly demonstrate impaired ribosome biogenesis, they suggest broader changes in RNA metabolism and raise the possibility that ribosome-related pathways may be affected. We also observed accumulation of rRNA in NCC protrusions of *Sh3Pxd2b*$^{nee−/−}$ mice, structures known to support local translation. Binding of ribosomal proteins to 5S rRNA is crucial for nuclear translocation of the 5S RNP complex and subsequent ribosome assembly (Pittman et al., 1999; Lin et al., 2001). One hypothesis is that impaired formation of cell protrusions in *Sh3Pxd2b*$^{nee−/−}$ mice may perturb local translation, leading to

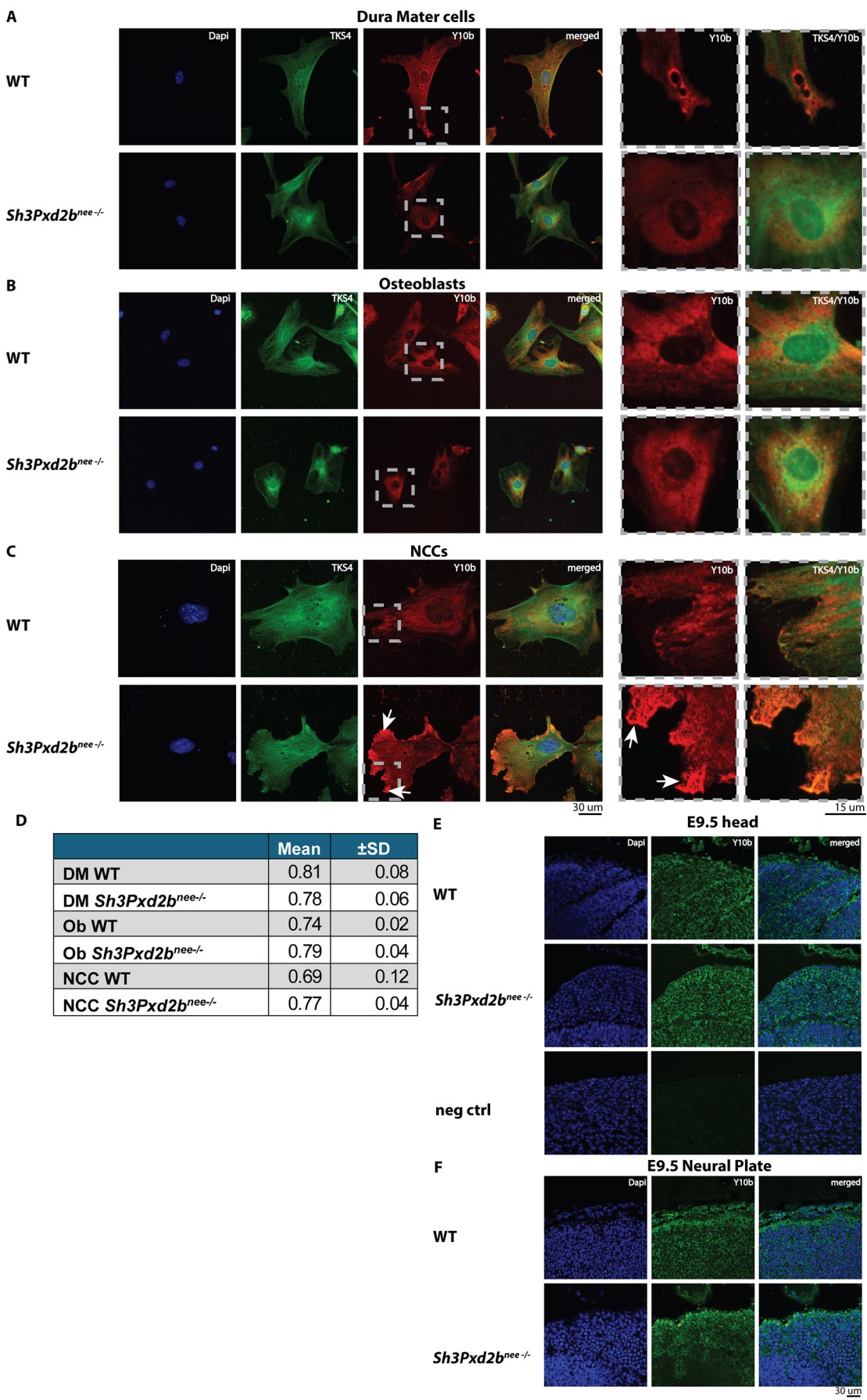

**Fig. 7.** See next page for legend.

**Fig. 7. Accumulation of ribosomal RNA in cell protrusions of Sh3Pxd2b$^{nee-/-}$ mice.** (A-C) Immunofluorescence staining for Y10b, DAPI and TKS4 in dura mater cells, osteoblasts and NCCs from WT and Sh3Pxd2b$^{nee-/-}$ mice. Boxed areas showing selected regions of interest are shown at higher magnification on the right. Overlap of green and red fluorophores appears as yellow. (A) Homogeneous distribution of TKS4 and Y10b in WT dura mater cells with focal Y10b expression near circular dorsal ruffles of the plasma membrane and perinuclear accumulation in mutant cells. (B) Homogeneous cytoplasmic distribution of TKS4 and Y10b extending radially towards the cell membrane in WT cells compared to perinuclear accumulation in mutant cells. (C) Homogeneous cytoplasmic distribution of TKS4 and Y10b in WT and mutant cells with enhanced immunoreactivity for Y10b in cell protrusions of mutant cells (arrows). (D) Table showing Pearson's coefficient indicating the correlation of overlap of TKS4 and Y10b. Data shown as mean±s.d. of three images each. (E,F) Immunofluorescence staining of tissue sections of the presumptive head and neural plate (NP) of WT and Sh3Pxd2b$^{nee-/-}$ E9.5 embryos, showing enhanced expression in the head (E) and decreased signal in the sub-layers of the NP (F) of mutant mice. Negative control for Y10b without primary antibody is shown in the bottom panels of E. See Fig. S2A for anatomical landmarks.

altered rRNA localization or processing, which may be reflected on a transcriptomic level by downregulation of nuclear encoded ribosomal RNA. The observed overlap between TKS4 and the rRNA-binding marker Y10b suggests that TKS4 might play a role in rRNA dynamics. Interestingly, Y10b displays cell type-specific localization differences with reduced protrusional localization in mutant dura mater cells but increased accumulation in protrusions of mutant NCCs, suggesting that rRNA-associated dynamics during migration may differ between these cell types. Although further investigation is needed, these findings point to a potential impairment of ribosome-associated processes that may contribute to the observed craniofacial phenotype in Sh3Pxd2b$^{nee-/-}$ mice.

Taken together, our data indicate that FTHS is characterized by dysfunctional podosome formation leading to proliferative and migratory defects, particularly in neural crest-derived tissues, during embryonic development. Additionally, our findings raise the possibility that disruptions in ribosome-related processes may contribute to the craniofacial phenotype. However, further studies are required to clarify the nature and extent of these potential mechanisms.

## MATERIALS AND METHODS
### Resources
Detailed information on resources used in this study can be found in Table S4.

### Animal husbandry
Animals were cared for in accordance with the regulations by the Institutional Animal Care and the Use Committee of Stanford University (Protocol #APLAC_8397). Heterozygous Sh3Pxd2b$^{nee+/-}$ mice were purchased from The Jackson Laboratory (Mao et al., 2009). Homozygous animals were generated by breeding Sh3Pxd2b$^{nee+/-}$ females and males on a pure C57/BL6 background. Genotyping was performed by PCR analysis of genomic DNA, followed by Sanger sequencing to detect the point deletion, according to the protocol supplied by The Jackson Laboratory. Later, genotyping was performed by Transnetyx, Inc. (Cordova, TN, USA). Wnt1-Cre2 mice and mT/mG mice were purchased from The Jackson Laboratory (Lewis et al., 2013; Muzumdar et al., 2007; Dinsmore et al., 2022). Heterozygous and homozygous animals were generated by breeding on a pure C57/BL6 background. Heterozygous female Wnt1-Cre2 mice were bred with homozygous mT/mG mice to generate a Wnt1-Cre2$^{+/-}$;mT/mG$^{+/-}$ mouse strain. Wnt1-Cre2$^{+/-}$;mT/mG$^{+/-}$ mice were bred with Sh3Pxd2b$^{nee+/-}$ mice to generate Wnt1-Cre2$^{+/-}$;mT/mG$^{+/-}$;Sh3Pxd2b$^{nee+/-}$ mice. Female and male Wnt1-Cre2$^{+/-}$;mT/mG$^{+/-}$;Sh3Pxd2b$^{nee+/-}$ were bred together to generate Wnt1-Cre2$^{+/-}$;mT/mG$^{+/-}$;Sh3Pxd2b$^{nee-/-}$ animals. Genotyping was

performed by PCR analysis of genomic DNA according to the protocol supplied by The Jackson Laboratory and by Transnetyx, Inc. Littermates of the same sex (male and female) and E9.5, pN1, 2, 3, 4, 5, 8, 12 and 17, and postnatal months 1, 2 and 6 animals were used for experiments.

### Primary cell cultures
Mice skulls (pN15) were harvested and washed in 2× Penicillin/Streptomycin (Gibco, 15140-122). Periosteum and dura mater were carefully stripped from the skull. Dura mater membranes were collected and placed in 12-well plates in PBS on ice, followed by 10 min of incubation at 37°C in 0.05% trypsin. Digestion was neutralized by adding Dulbecco's modified Eagle's medium (DMEM), supplemented with 10% fetal bovine serum (FBS) and 1% Penicillin/Streptomycin (Li et al., 2007; Mehrara et al., 1999). Osteoblasts were harvested as previously described (Li et al., 2015). In brief, skulls were minced into small chips less than 1 mm in size, followed by digestion with 0.2% Dispase and 0.1% Collagenase A in medium. Digestion was repeated four times, each for 20 min. Digestions were then pooled and neutralized with Alpha MEM, supplemented with 10% FBS and 1% Penicillin/Streptomycin, pelleted, resuspended in growth medium and plated. Cells were incubated at 37°C in 5% $CO_2$. Passage 0-3 cells were used for experiments.

Cranial neural crest cells were isolated as previously described (Gonzalez Malagon et al., 2019; Moore and Trainor, 2022). In brief, tissue culture plates were harvested with 1 μg/ml fibronectin in Dulbecco's PBS. E9.5 embryos were utilized. The uterus was dissected into ice-cold PBS and embryos separated. Decidual tissue, Reichert's membrane, yolk sac and amnion were carefully removed. The NP was visualized and isolated just above the heart. Underlying tissue was trimmed away. For migration assays, the tissue was plated at 37°C in 5% $CO_2$ until attached, or for at least 1 h, before neural crest basal medium (DMEM/F12 GlutaMAX, 1% Penicillin/Streptomycin, 15% FBS) was added. For immunofluorescence, the isolated NP was further digested in 1 mg/ml Dispase II (Roche Holding AG) for 5 min at room temperature and washed three times with Tyrode's media (Sigma-Aldrich) before being plated. Sex of cell lines was not considered and cell lines were not authenticated.

### Tissue harvesting and processing
Animals were euthanized on pN1, 2, 3, 4, 5, 8, 12 and 17 and postnatal months 1, 2 and 6. For each time point, at least three animals were euthanized and processed for histology and whole-mount staining. Animals were asphyxiated by $CO_2$ and the skull was harvested. For histology, skulls were briefly washed in PBS, fixed in 4% paraformaldehyde (PFA) overnight at 4°C and decalcified in 19% EDTA for the appropriate time. Specimens were then dehydrated in 10% and 20% sucrose (each 1 h), followed by 30% sucrose incubation at 4°C overnight. Skulls were then cryo pre-embedded in optimal cutting temperature compound (OCT-Tissue-Tek) at −80°C and −20°C (both overnight), thawed and re-embedded accordingly in the right orientation.

Embryonic specimens were collected at E9.5 for analysis. At least three embryos were harvested and processed for histology. Pregnant females were asphyxiated by $CO_2$, the uterus dissected into ice-cold PBS and individual embryos separated. Decidual tissue, Reichert's membrane, yolk sac and amnion were removed. The embryo was briefly washed in PBS, fixed in 4% PFA for 1 h at 4°C and transferred into 30% sucrose until the specimen sunk to the bottom of the dish. Specimen were then cryo-embedded in 7.5% gelatin in 10% sucrose and stored at −80°C.

### Histology
Skull specimens were sectioned in an anterior-posterior direction (coronal plane) at 10 μm thickness. Depending on size, 450-600 cryosections were obtained per animal. Embryonic samples were sectioned in a medial-lateral direction (sagittal plane) at 7 μm thickness. Representative slides for each area were stained by Movat's Pentachrome, Hematoxylin and Eosin or TRAP according to standard procedures or processed for immunofluorescence. For TRAP staining, frozen sections were fixed briefly in chilled 10% neutral buffered formalin, rehydrated in PBS and rinsed in distilled water. Slides were incubated in pre-warmed TRAP staining solution at 37°C for ~30 min, rinsed, and counterstained with Fast Green. After air-drying, slides were cleared in xylene and mounted. Images were acquired using either a Leica inverted

microscope DMI 4000B and Leica Application Suite (LAS X) or a digital slide scanner (Motic EasyScan One) and processed using Adobe Photoshop. For larger fields of view, images were tiled either automatically by the scanner or manually when required.

## Whole-mount staining

Alizarin Red whole-mount staining was performed according to an adapted protocol described by Rigueur and Lyons (2014). Briefly, after harvesting, skulls were cleaned in PBS and ACK (Ammonium-Chloride-Potassium) Lysing Buffer and fixed in a graded ethanol dilution (20%, 50%, 70%, 80% each 1 h, 95% overnight) at 4°C. Specimens were incubated in Alizarin Red solution (10 mg in 200 ml 1% KOH) for 3-4 h at room temperature. This was followed by a clearing procedure utilizing graded glycerol (20% in 1% KOH, 20% in $H_2O$, 50% in $H_2O$, 70% in $H_2O$, storage in 95% in $H_2O$).

For calcein whole-mount staining, skulls were incubated for 24 h at 37°C in Alpha Minimum Essential Medium (MEM) supplemented with 10% FBS, 1% Penicillin/Streptomycin and 1 μm/ml calcein.

Skulls were analyzed and imaged using a Leica M205FA Fluorescence stereo and LAS X and processed utilizing Adobe Photoshop. Morphometric analysis was performed on whole mounts using the open-source Fiji/ImageJ software.

## *In vitro* assays

For proliferation, cells were seeded in 6-well plates and, when confluent, labeled using the Click-iT™ EdU Cell Proliferation Kit for Imaging with Alexa Fluor™ 488 dye (Thermo Fisher Scientific Inc.) according to the manufacturer's instructions. The percentage of EdU-positive cells was quantified using open-source Fiji/ImageJ software. For migration assay, cells were seeded in 6-well plates. Upon confluency, a scratch was performed using a 200 µl pipette tip. An image of the migration area was taken at the start of the experiment (0 h), and at 7, 24, 34, 48, 62 and 72 h or until the migration area was fully covered with cells. The size of the migration area was determined using ImageJ and divided by the size of the migration area at the initial start of the assay to determine percentage of migration area. For migration of neural crest cells, an image was taken after plating of neural tissue and at 24, 48 and 72 h. The size of the migration was determined using ImageJ and divided by the size of the tissue at plating. Apoptosis was assessed by TUNEL assay using the *In Situ* Cell death Detection Kit, Fluorescein (Roche), according to the manufacturer's instructions. Images were using a Leica DMI 4000B inverted microscope system and Leica Application Suite (LAS X, Leica Microsystems) and processed utilizing Adobe Photoshop (Adobe Inc.). For the osteogenic assays, cells were seeded in 12-well plates and cultured in osteogenic medium (Alpha MEM, 10% FBS, 1% Penicillin/Streptomycin, 10 mM β-glycerophosphate and 100 µg/ml ascorbic acid) for 3 weeks or until matrix mineralization was observed. For quantification, cells were fixed with 100% ethanol for 15 min and incubated with Alizarin Red stain for 1 h. Fixed cells were left to dry, followed by incubation with 20% methanol and 10% acetic acid in $dH_2O$ for 15 min to quantify Alizarin Red using a spectrophotometer at 450 nm wavelength (NanoDrop™ One/One$^C$ Microvolume UV-Vis Spectrophotometer).

## Animal surgeries

Animal surgeries were performed in accordance with the Stanford University Animal Care and Use Committee Guidelines (Protocol #APLAC_8397). Animals underwent calvarial defect surgery as previously described (Behr et al., 2010b). After anesthesia, an incision was made along the midline of the skull. The pericranium was carefully removed and 2 mm calvarial defects were created in the right parietal bone using a trephine drill taking care not to damage the underlying dura mater. The incision was closed using a 6-0 nylon suture. Animals were either monitored to assess bone regeneration by micro-CT scanning at postoperative day 1 (pN22) and 3, 6 and 18 weeks postoperatively, or skulls were harvested and processed for histology, as described above.

## Micro-CT scanning

Animals were anesthetized by 2% isoflurane before scanning. Scans were performed using a high-resolution Bruker SkyScan 1276 CMOS Micro CT. Reconstruction was performed with NRecon software (Bruker Corporation)

and 3D images produced using CTAn and CTVol (Bruker Corporation). The remaining defect area was determined using the 'Magic Wand' tool in Adobe Photoshop (Adobe Inc.) and divided by the mean of the defect size on the first postoperative day to determine percentage healing.

## Immunofluorescence/immunocytochemistry

For immunocytochemistry, if applicable, to induce migration, cells were stimulated with TGFβ1 at 25 ng/ml for 5 h (Murphy et al., 2011). For immunofluorescence staining, slides with tissue sections or cells were allowed to equilibrate at room temperature for 10 min, washed in PBS and fixed with 4% PFA or methanol for 20 min. After fixation, slides were washed in PBST (0.05% Tween-20 in PBS) and blocked for 1 h with 10% normal donkey serum in PBST. Slides were incubated with primary antibodies against SOX9 (1:250), COL2A1 (1:50), COL10A1 (1:100), BGLAP (osteocalcin) (1:500), SH3PXD2B (TKS4) (1:250), SOX10 (1:200), PCNA (1:200) or ribosomal RNA/Y10b (1:50) in 2% donkey serum in PBST overnight according to the manufacturer's instructions. Following incubation, slides were washed in PBST and incubated with a goat anti-rabbit or goat anti-mouse fluorescent secondary antibody (Alexa Fluor 488 or 647; 1:1000; see Table S4) for detection. If applicable, slides were washed and incubated with fluorescent Phalloidin (F-actin) according to the manufacturer's instructions. Negative controls were performed with no primary antibody. Slides were mounted with Fluoromount-G with DAPI. Laser-scanning confocal microscopy was performed with a Zeiss LSM880 or Leica SP8 microscope located in the Stanford University Cell Sciences Imaging Core Facility (RRID:SCR_017787).

Colocalization analysis was performed using 'Just Another Colocalization Plugin' (JACoB) in ImageJ. For each fluorophore, 8-bit images were used to calculate the Pearson's coefficient (r) quantifying the degree of overlap between channels. The Pearson's coefficient ranges from +1, indicating positive correlation, to 0, indicating no correlation, to −1, indicating negative correlation. Results were based on three images each, with coefficients averaged across the samples (Bolte and Cordelières, 2006).

For identification of podosome formation, immunofluorescence staining was performed as previously described (Moreau et al., 2003). In brief, cytoskeletal buffer (CB) was prepared in Milli-Q water supplemented with 10 mM morpholineethanesulfonic acid, 150 mM NaCl, 5 mM EGTA, 5 mM $MgCl_2$ and 5 mM glucose at pH 6.1. Cells were stimulated with 25 ng/ml TGFβ1 for 5 h or left unstimulated (Murphy et al., 2011). Cells were fixed in 3% PFA in CB for 10 min at room temperature and permeabilized with 0.1% Triton X-100 for 1 min, followed by three washes with CB. Cells were incubated in blocking solution (1% bovine serum albumin, 2% FBS in PBS) for 30 min at room temperature. Primary antibody against cortactin (1:500 in blocking solution) was added for 30 min at room temperature followed by three washes with TBS (20 mM Tris, 150 mM NaCl, 2 mM EGTA, 2 mM $MgCl_2$ at pH 7.5). Goat anti-rabbit fluorescent secondary antibody (1:2000 in blocking solution; Table S4) was added for 30 min, followed by three washes with TBS. Cells were incubated with Phalloidin according to the manufacturers' instructions for 45 min and washed twice with PBS before mounting.

## Bulk RNA sequencing

Skulls were harvested at pN15 and immediately immersed in cold TRIzol Reagent on ice followed by RNA extraction. A total of six skulls (three WT mouse skulls and three *Sh3Pxd2b*$^{nee−/−}$ mouse skulls) were harvested. RNA extraction and reverse transcription were performed as previously described (Quarto et al., 2018). Bulk RNA sequencing was performed by the Protein and Nucleic Acid (PAN) Facility at Stanford University. Libraries were prepared using a poly(A) enrichment protocol. Sequencing data in FASTQ format were aligned in HISAT2 and assembled in String Tie using the mm10 reference genome with default settings (e.g. minimum isoform fraction >0.15, minimum assembled transcript length >200) (Kim et al., 2019; Pertea et al., 2015). The resulting sample-level feature matrices were then ported to edgeR, filtered for lowly expressed genes, normalized by trimmed mean of M values (TMM), dispersion-estimated, and log-transformed using the standard package pipeline (Robinson et al., 2010; Chen et al., 2025). Genewise exact tests were applied using the built-in EdgeR functions to identify differentially expressed genes between the mutant and WT groups, including a false discovery rate-adjusted *P*-value threshold of <0.05 for downstream analysis. Heatmaps were created based

on log2-normalized expression. Gene set enrichment analysis was performed on ranked lists of differentially expressed genes using EnrichR v2.1. (Chen et al., 2013; Kuleshov et al., 2016; Xie et al., 2021) and KEGG 2021 mouse pathways database. Principal component analysis was used to analyze aggregate separation between WT and mutant mouse skull transcriptomes. Open source software Gephi and ToppCluster (Bastian et al., 2009; Kaimal et al., 2010) were used to perform network analysis.

## Agilent Bioanalyzer small RNA analysis

Total RNA was extracted from whole-mount skulls at pN15 using TRIzol Reagent followed by column-based purification. RNA from one skull per condition was used for each analysis. Small RNA profiles were assessed using the Agilent 2100 Bioanalyzer system with the Small RNA assay according to the manufacturer's instructions. The analysis was performed by the Protein and Nucleic Acid (PAN) Facility at Stanford University. MicroRNAs typically appear in the 10-40 nucleotide range, 5S rRNA is generally detected between 84 and 96 nucleotides and 5.8S rRNA between 135 and 150 nucleotides on the small RNA chip. The ratios were calculated by dividing the concentration obtained for microRNA, 5S rRNA and 5.8S rRNA, respectively, by the total small RNA concentration (Agilent Technologies, Publication Number 5989-8539EN; Bir et al., 2024).

## Statistical analysis

Unpaired, two-tailed Student's $t$-test was performed to test for differences. ANOVA followed by Šidák's multiple comparisons test was performed to assess differences between groups. $P<0.05$ was considered significant. GraphPad Prism was used to create graphs. Detailed information regarding type of statistical test and number of biological replicates is provided in figure legends.

## Acknowledgements
The Leica SP8 confocal microscope was funded, in apart, by Award Number 1S10OD010580 from the National Center for Research Resources (NCRR).

## Competing interests
The authors declare no competing or financial interests.

## Author contributions
Conceptualization: J.H., S.M.; Data curation: J.H., S.M., M.L.-T., J.L.G.; Formal analysis: J.H., J.L.G.; Funding acquisition: J.H., M.T.L.; Investigation: J.H., S.M., M.L.-T.; Methodology: J.H., S.M.; Project administration: M.T.L.; Resources: M.T.L.; Visualization: J.H., S.M.; Writing – original draft: J.H.; Writing – review & editing: J.H., S.M., M.L.-T., J.L.G., M.T.L.

## Funding
This work was supported by the German Research Foundation (Deutsche Forschungsgemeinschaft; HU 2817/1-1 to J.H.), the Transplant and Tissue Engineering Center of Excellence leadership group from the Stanford Children's Hospital (Lucile Packard Children's Hospital fellowship for J.H.), the National Institutes of Health (U24DE029463 and R01DE027323 to M.T.L.; F32-HL167318 to J.L.G.), the Hagey Laboratory for Pediatric Regenerative Medicine (Stanford University School of Medicine) and the Wu Tsai Human Performance Alliance (Stanford University). Open Access funding provided by Stanford University. Deposited in PMC for immediate release.

## Data and resource availability
RNA sequencing data have been deposited in NCBI's Gene Expression Omnibus and are accessible through GEO Series accession number GSE283030. All other relevant data and details of resources can be found within the article and its supplementary information.

## Peer review history
The peer review history is available online at https://journals.biologists.com/dev/lookup/doi/10.1242/dev.204631.reviewer-comments.pdf

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
