## [Peer Review File · Development (Cambridge, England)]

The *Sh3Pxd2b^{nee-/-}* mouse reveals developmental features of Frank-ter Haar Syndrome

Julika Huber, Siddharth Menon, Michael Lopez-Torres, Jason L. Guo and Michael T. Longaker

DOI: 10.1242/dev.204631

Editor: James Briscoe

Review timeline

Original submission:	3 January 2025
Editorial decision:	13 February 2025
First revision received:	27 June 2025
Editorial decision:	12 September 2025
Second revision received:	23 November 2025
Accepted:	16 December 2025

Original submission

First decision letter

MS ID#: dev.204631

MS TITLE: The *Sh3Pxd2b^{nee-/-}* mouse: an attractive model unveiling novel developmental features in Frank-ter-Haar Syndrome

AUTHORS: Julika Huber, Siddharth Menon, Michael Lopez-Torres, Jason L. Guo and Michael T. Longaker

Dear Dr Huber,

I have now received all the referees' reports on the above manuscript, and have reached a decision. The referees' comments are appended below, or you can access them online: please go to:

As you will see, the referees express interest in your work, but have some significant criticisms and recommend a substantial revision of your manuscript before we can consider publication. One issue concerns image presentation and the cropping of skull images. The microCT, alizarin red, and calcein staining images appear to omit crucial anatomical features that hampers interpretation and comparison. Similarly, sections from E9.5 embryos lack essential anatomical landmarks needed to verify consistent positioning. Beyond imaging issues, the referees raise questions about antibody specificity, this requires thorough investigation and validation. The RNA sequencing analysis at postnatal day 15 needs more detailed methodology and stronger experimental support for its conclusions about ribosome biogenesis. I would also encourage you to address the interpretation of the neural crest-specific migration defects. Additional quantification would strengthen several key findings, including proliferation assays and protein localisation patterns. Overall, more experimental support is need to support the claim of ribosomal involvement in Frank-ter-Haar Syndrome.

If you are able to revise the manuscript along the lines suggested, which may involve further experiments, I will be happy to receive a revised version of the manuscript. Your revised paper will be re-reviewed by one or more of the original referees, and acceptance of your manuscript will depend on your addressing satisfactorily the reviewers' major concerns. Please also note that

Development will normally permit only one round of major revision. If it would be helpful, you are welcome to contact us to discuss your revision in greater detail. Please send us a point-by-point response indicating your plans for addressing the referees' comments, and we will look over this and provide further guidance.

Please attend to all of the reviewers' comments and ensure that you clearly highlight all changes made in the revised manuscript. Please avoid using 'Tracked changes' in Word files as these are lost in PDF conversion. I should be grateful if you would also provide a point-by-point response detailing how you have dealt with the points raised by the reviewers in the 'Response to Reviewers' box. If you do not agree with any of their criticisms or suggestions please explain clearly why this is so.

Reviewer 1

Advance summary and potential significance to field

In their manuscript with the title "The Sh3Pxd2bnee^{-/-} mouse: an attractive model unveiling novel developmental features in Frank-ter-Haar Syndrome", Huber et al. analyze craniofacial defects in a mouse model of a rare human disease characterized by skeletal abnormalities and developmental delay. While a first description of phenotypes in this mouse model was already published about 15 years ago (PMID: 20137777), the current study provides a much more detailed picture of the defects in the cranial sutures. Moreover, Huber et al. have used in vitro experiments with cultured dura mater cells, osteoblasts and neural crest cells together with RNA sequencing to obtain a better insight into the cellular and molecular alterations in Sh3Pxd2bnee^{-/-} mutants. Overall, this is an interesting study that offers new insights into a poorly understood syndrome, but there is a couple of critical issues that needs to be addressed prior to publication:

Major points:

It is very puzzling that there is still anti-TKS4 immunostaining in the mutant. Does this mean that this is not a null allele so that the mutants might still produce a truncated, potentially partially functional gene product? Obviously, this question is absolutely central and needs to be resolved.

In the same context, the authors should provide results validating the anti-TKS4 antibody and immunostainings. The substantial signal in ^{-/-} tissues and cells raises concerns regarding the specificity of the staining and the impact of potential background signal.

The nature or cellular identity of the postnatal dura mater cells is not entirely clear? Is this a mixture of different cell types? Along the same lines, it would be very useful to clarify in the text whether the contribution of dura mater-derived skull progenitor cells is confined to embryonic development or continues postnatally.

In line 278, it is stated that Y10B is "marker for ribosomal rRNAs", but Y10B is actually not a marker but a monoclonal antibody detecting rRNA. Regarding the proposed "colocalization" of TKS4 and rRNA, this conclusion is not sufficiently supported by data. Apart from the serious issue with the specificity of the anti-TKS4 immunostaining, the techniques used in the study cannot demonstrate actual colocalization and it might be simply that signals (if specific) are partially overlapping. Along the same lines, there is simply not enough evidence that ribosome biogenesis is altered in Sh3Pxd2bnee^{-/-} mutants. This conclusion would require a much more detailed analysis of ribosomes and rRNA beyond staining with a single antibody. The authors should rephrase/eliminate the relevant statements or provide additional results that directly address ribosome biogenesis and function in the mutant cells.

Minor points:

In line 168, it is stated that WT samples show "increased proliferation compared to mutant mice". As WT proliferation is not increased but at its natural level, it would be better to write that mutants show lower proliferation than WT controls.

In line 186-188, the authors write "The percentage of cells with podosome formation was significantly decreased in TGF β 187 stimulated WT dura mater cells compared to TGF β -stimulated mutant dura mater cells". Again, this statement should be reversed because the WT is obviously the natural/control level, but the statement also appears to contradict the actual data in Fig. 4, which shows persistently lower podosomes in -/- cells. Please clarify and correct as required.

Please improve the writing in lines 190 to 207. It is currently not easy to follow the description of the results in this paragraph.

Reviewer 2

Advance summary and potential significance to field

The manuscript by Huber et al. uses a previously established Sh3Pxd2b (nee) mutant model to explore the mechanisms underlying Frank-ter-Haar syndrome (FTHS). This study extends previous analyses of this mouse model, providing an in-depth characterization of the calvarial phenotype. The study uses a combination of microCT imaging, histology and in vitro assays to demonstrate reduced osteogenesis in the Sh3Pxd2b (nee) mouse model. The study further uses in vitro assays to show reduced cell migration and reduced proliferation in the Sh3Pxd2b-/- mice. There are significant issues with the image quality for some of these studies, making it difficult to interpret the data. The last part of the study uses RNA-seq to explore potential molecular mechanisms, and explores downregulation of ribosome biogenesis; however, I have significant concerns regarding the interpretation of this data, which is detailed below. Overall, this study provides a more comprehensive picture of the Sh3Pxd2b-/- phenotype and has potential to provide mechanistic information if additional studies are performed.

Comments for the author

1. The skull is represented in multiple images in Figure 1 using microCT (B), alizarin red (C), and calcein staining (D), and histology (E). These images appear to have been cropped so that portions of the skull are removed from the images, making it difficult to interpret and compare the images. The zygomatic is missing from the microCT images in both control and mutant in panel B. The nasal bone is largely missing from all images in panels C and D in addition to portions of the calvarial bones. Including the full images would improve the quality of the figure as well as making clear the interpretation of the data.
2. The images in Figure 1 are very small and difficult to interpret. Please consider moving some of this data to supplementary figures so that the images can be presented clearly.
3. The figure legends state experiments were performed in triplicate with n=3 per group, but it is not clear whether these were littermates, and whether littermates were sex matched. Please more clearly indicate this information.
4. In Figure 2, since the wt mice do not show much bone regeneration, this could suggest increased osteoclast activity. Is there any evidence for increased osteoclast activity, for example TRAP staining? And if there is an imbalance in bone homeostasis, does this contribute to the suture patency over time? Could the suture phenotypes be a combination of both reduced osteogenesis and increased osteoclast activity?
5. The localization of the antibody stain in Figure 3 would benefit from additional explanation. Given the nee mutation leads to a truncated protein, this seems related to the work of Adam et al., 2015 that showing changes in localization of Tks4 truncated protein, as well as localization to aggresomes, which would be worth citing. The images in the osteoblasts do not seem to show any differences between control and mutant cells with respect to the Tks4 localization; and it seems that there are more perinuclear condensates in wt versus mutants in the images in the figure. Given that the conclusion was mutants were more condensed, is there quantification to demonstrate this? What percentage of cells shows the localization pattern represented?
6. All sections from E9.5 embryos need anatomical landmarks (Figs 3, 4, 5 and 7) to show the sections are from the same location in the embryo. It is unclear which specific region "head area" refers to in Figure 3. Showing the entire section with anatomical landmarks would be beneficial to orient readers to where in embryo the expression pattern was examined.
7. The DAPI in Fig. 3 panel C seems to show different orientation and the nuclei appear larger in the mutant image- were these images taken at the same magnification? The staining for Tks4

also does not seem to be very specific in these sections - is it possible that there was just high background? At E9.5, the staining looks reduced in wt compared to mutant, in contrast to the statement of no difference.

8. Is the noted difference in Tks4 expression in Fig. 3D a true reduction in the mutant, or just a lack of those cell populations? The differences stated in the text about Fig. 3 are not easily distinguished and could benefit from image analysis to support the conclusions.

9. The EdU proliferation assay shows a change in vitro, but additional quantification would be beneficial on the sections from E9.5 and pN3. In Fig. 4D, what do "outer tissue layers" refer to? The morphology of the tissue sections between control and mutant look different here, so are differences in signal due to a different area of the embryo? Including the entire section for anatomical reference would be beneficial.

10. In Fig. 4E and F, is the noted increase in proliferation in the mesenchyme in wt simply because there are more cells in wts? Quantification of the percentage of PCNA+ cells/DAPI would be informative here since the morphology between controls and mutants is already quite different.

11. TGFB treatment did not appear to increase podosome formation in controls (data in Fig. 4I). Is there an explanation for this?

12. The images in Fig. 4M, N need to represent the full section. As the data is currently represented, it is not possible to determine if there is a reduction in migration or if a portion of the image is simply missing. The conclusion that migration is more affected in cells of neural crest cell origin requires additional support or explanation. It is not clear from the materials and methods that the osteoblasts are solely those of neural crest cell origin, or whether they are a mixed population of neural crest and mesoderm derived, or only mesoderm. If mesoderm versus neural crest derived osteoblasts were examined, please include that information. The images in Fig. 4 M, N are not clear enough to fully interpret, but it would appear that neither mesoderm nor neural crest derived cells are migrating.

13. Fig. 5C is a repeat of what was already in Fig. 4D, and Fig 5B is a repeat of Fig. 3C. It seems more logical to have this data just once, and having this younger data altogether in Fig. 5 makes sense. Please provide anatomical information for the sections.

14. The TUNEL staining looks increased in the mutants in Fig. 5D - was this true across multiple sections from multiple embryos? The text states that TUNEL is not changed.

15. The explant migration assays are informative for examining podosomes, but is there evidence of a neural crest cell migration defect in vivo? Does the overall migration pattern of neural crest cells appear abnormal at E9.5 or E10.5 in the mutant embryos with Wnt1Cre;mT/mG? When do differences in the craniofacial morphology appear in the Sh3Pxd2b^{-/-} mice? This is an important point regarding mechanism. If there are already changes in proliferation and migration at E9.5 that lead to the differences observed postnatally, then this would be important to examine in greater detail.

16. The RNA-seq experiment was performed at pN15 on whole skulls (mesoderm and neural crest origin combined). This time point is well after defects in the mutant mice are apparent, better reflecting the role for Sh2Pxd2b during postnatal growth. Since the previous section seemed to be suggesting that Sh3Pxd2b had an early role in neural crest cell development, could you provide some rationale for choosing pN15?

17. Please include more information about the RNA-seq experiment in the materials and methods section with details on the library preparation, quality control cutoffs, and analyses. For example, was there a minimum expression threshold used (ie TPM>2)? Was p value or adjusted p value used for cutoff (text states p and not adjusted p, which should be used to account for multiple testing)?

If the library was made using polyA selection, differences in ribosomal RNA and other non-coding RNAs should be interpreted with caution and carefully validated. If a significant difference is seen in 5S rRNA in the mutant versus control sample using an assay like the Agilent Bioanalyzer or Northern blot, then this would support a true change versus an artifact from the library preparation method. For the KEGG analysis, please include the table on the effect size and direction - are all of the genes presented in Fig. 7E downregulated, or were some upregulated?

18. The upregulated genes mentioned that were metabolic and biochemical could still be of relevance in this system, especially given potential changes in bone homeostasis (Fig 2). This point could be worth further consideration in the discussion.

19. The Y10b antibody staining seems to suggest that rRNA is not downregulated in mutants relative to controls, in contrast to the RNA-seq result. Can you explain this? (However, it should be noted that Y10b is not a marker of 5S rRNA specifically, just rRNA more broadly.)

20. There is no experimental evidence provided to support downregulation of ribosome biogenesis. Including experiments to show downregulation of ribosome biogenesis in the *Sh3Pxd2b* mutant embryos would provide preliminary support for the model proposed in Fig. 7G.

21. The colocalization of Y10b and Tks4 is perhaps not surprising given both are broadly expressed. This is correlative, and further evidence is necessary to determine if this is meaningful, which should be highlighted in the discussion. The suggestion of FTHS as a ribosomopathy should be avoided until there is definitive experimental evidence supporting diminished ribosome biogenesis.

22. There is a point in the discussion about migration defects specifically in neural crest cell derived tissues, but it seems likely that there are also these migration defects in mesoderm-derived calvarial bones based on the data presented in Fig. 4.

Minor points

- There is aberrant text over a couple of panels in Figure 1 part F; and "merged" is cut off on the bottom portion of the text
- Mouse nomenclature for protein is Tks4 (only the first letter capitalized)
- Spell out EdU the first time and then use the abbreviation.

First revision

Author response to reviewers' comments

Reviewer 1: In their manuscript with the title "The *Sh3Pxd2b^{nee} -/-* mouse: an attractive model unveiling novel developmental features in Frank-ter-Haar Syndrome", Huber et al. analyze craniofacial defects in a mouse model of a rare human disease characterized by skeletal abnormalities and developmental delay. While a first description of phenotypes in this mouse model was already published about 15 years ago (PMID: 20137777), the current study provides a much more detailed picture of the defects in the cranial sutures. Moreover, Huber et al. have used in vitro experiments with cultured dura mater cells, osteoblasts and neural crest cells together with RNA sequencing to obtain a better insight into the cellular and molecular alterations in *Sh3Pxd2b^{nee} -/-* mutants. Overall, this is an interesting study that offers new insights into a poorly understood syndrome, but there is a couple of critical issues that needs to be addressed prior to publication:

Major points:

It is very puzzling that there is still anti-TKS4 immunostaining in the mutant. Does this mean that this is not a null allele so that the mutants might still produce a truncated, potentially partially functional gene product? Obviously, this question is absolutely central and needs to be resolved.

Author's reply: We thank you for your comment and we agree that this question is absolutely central. The *Sh3Pxd2b^{nee} -/-* mouse harbors a spontaneous single base pair deletion in the last exon of the *Sh3Pxd2b* gene on chromosome 11. This frame shift mutation indeed causes a protein truncation, which alters a part of the third SH3 domain and deletes the fourth SH3 domain (Mao et al., 2009). Due to the type of mutation in the *Sh3Pxd2b^{nee} -/-* mouse, the level of TKS4 protein should be unaffected compared to WT mice, however the protein structure is altered and therefore the protein is not fully functional. Thus, we do expect to see anti-TKS4 immunostaining in the mutant mouse. We have clarified the nature of the mutation on Page 4, lines 81-88.

In the same context, the authors should provide results validating the anti-TKS4 antibody and immunostainings. The substantial signal in *-/-* tissues and cells raises concerns regarding the specificity of the staining and the impact of potential background signal.

Author's reply: We appreciate the reviewer's concerns regarding antibody specificity. To rule out non-specific binding of the secondary antibody, we included a no-primary antibody control, which showed no detectable signal in Figure 3. TKS proteins are ubiquitously expressed in tissues (with the exception of the spleen for TKS4) (Kudlik et al., 2020), thus we consider the WT tissues as a

positive control. Given that the TKS4 protein in the *Sh3Pxd2b^{nee} -/-* mouse is truncated, we fully expected positive immunostaining for TKS4 in mutant tissues. The staining pattern observed in WT and mutant tissues is consistent with descriptions in previous studies (Bögel et al., 2012, Buschman et al., 2009)

The nature or cellular identity of the postnatal dura mater cells is not entirely clear? Is this a mixture of different cell types? Along the same lines, it would be very useful to clarify in the text whether the contribution of dura mater-derived skull progenitor cells is confined to embryonic development or continues postnatally.

Author's reply: Thank you for your comment. Dura mater cells are of neural crest origin (Levi et al., 2011, Jiang et al., 2002). Dura mater cells are harvested from the dura mater underlying the skull bones and have previously been shown to coexpress desmoplakin and vimentin, known markers for mesenchymal-like meningeal cells. The method we used to isolate and study dura mater cells was first established in our lab in 1999 (Mehrrara et al., 1999) and has since been widely applied in investigations of cranial suture biology and osteogenesis. This original reference has been added in the methods sections on page 18, line 455-456. While our isolation method enriches for mesenchymal-like meningeal cells, it is important to note that the postnatal dura mater is a heterogeneous tissue that may include other cell types, which should be taken into consideration when interpreting results.

While dura mater-derived progenitor cells play a critical role during embryonic development (Jiang et al., 2002), our study, as well as previous work, demonstrates that these cells continue to contribute to skull development postnatally, including suture fusion and calvarial reossification (Hobar et al., 1993, Mossaz and Kokich, 1981, Yu et al., 1997, Spector et al., 2002, Bradley et al., 1997, Greenwald et al., 2000). We have added this clarification to the manuscript on page 7, lines 167-178.

In line 278, it is stated that Y10B is "marker for ribosomal rRNAs", but Y10B is actually not a marker but a monoclonal antibody detecting rRNA.

Author's reply: We thank you for your comment. We have changed the wording now on page 13, line 322-323.

Regarding the proposed "colocalization" of TKS4 and rRNA, this conclusion is not sufficiently supported by data. Apart from the serious issue with the specificity of the anti-TKS4 immunostaining, the techniques used in the study cannot demonstrate actual colocalization and it might be simply that signals (if specific) are partially overlapping.

Author's reply: We thank the reviewer for this important observation. Our analysis is based on fluorescence intensity correlation using Pearson's coefficient, which does not provide spatial resolution and therefore cannot definitely confirm molecular colocalization. In response, we have revised the relevant sections of the manuscript to use the word "overlap" rather than "colocalization" (lines 339-343, 354-355, 439-440, Figure 7 legend).

Along the same lines, there is simply not enough evidence that ribosome biogenesis is altered in *Sh3Pxd2b^{nee} -/-* mutants. This conclusion would require a much more detailed analysis of ribosomes and rRNA beyond staining with a single antibody. The authors should rephrase/eliminate the relevant statements or provide additional results that directly address ribosome biogenesis and function in the mutant cells.

Author's reply: We thank the reviewer for this important and fully agree that a more detailed analysis would be required to definitively demonstrate alterations in ribosome biogenesis. In the revised manuscript, we have carefully rephrased relevant sections to reflect this, using more cautious language such as "suggestive of" or "potential involvement of ribosome-related processes," and we clearly state that further studies are needed to substantiate this hypothesis.

To further support our observations and following a suggestion from Reviewer 2 in comment #17, we have now included additional data from Agilent Bioanalyzer analysis of RNA extracted

from skull tissue, which revealed a reduction in both small RNA concentration and the proportion of microRNAs in mutant mice compared to WT. Notably, the 5S rRNA peak also appeared consistently diminished in mutant samples. While these findings are not sufficient to confirm a defect in ribosome biogenesis, they align with the transcriptomic downregulation of ribosome-related genes and the altered subcellular localization of rRNA observed by Y10b staining. Together, these independent datasets suggest that ribosome-associated processes may be disrupted in mutant cells, though additional functional studies will be necessary to directly assess ribosome biogenesis and activity.

These results have updated the relevant text throughout the abstract, results (line 309-313, 349-355) and discussion (line 427-443, 446-449) sections to reflect these clarifications and to avoid overinterpretation of current data.

Minor points:

In line 168, it is stated that WT samples show "increased proliferation compared to mutant mice". As WT proliferation is not increased but at its natural level, it would be better to write that mutants show lower proliferation than WT controls.

Author's reply: We thank you for your comment and have revised the sentence according to the reviewers' suggestions, now in line 208-211.

In line 186-188, the authors write "The percentage of cells with podosome formation was significantly decreased in TGFβ187 stimulated WT dura mater cells compared to TGFβ- stimulated mutant dura mater cells". Again, this statement should be reversed because the WT is obviously the natural/control level, but the statement also appears to contradict the actual data in Fig. 4, which shows persistently lower podosomes in -/- cells. Please clarify and correct as required.

Author's reply: We thank the reviewer for pointing out the error. The statement was indeed incorrect and did not reflect the data shown in Figure 4I. We have revised the sentence to accurately reflect the results, which show a significantly lower percentage of podosome-forming cells in mutant dura mater cells compared to WT following TGFβ-stimulation. The text has been revised in line 229.

Please improve the writing in lines 190 to 207. It is currently not easy to follow the description of the results in this paragraph.

Author's reply: We thank you for your comment and have rephrased the paragraph with clearer more succinct language (lines 232-248).

Reviewer 2: SUMMARY OF THE ADVANCE MADE IN THIS PAPER AND ITS POTENTIAL SIGNIFICANCE TO THE FIELD

The manuscript by Huber et al. uses a previously established Sh3Pxd2b (nee) mutant model to explore the mechanisms underlying Frank-ter-Haar syndrome (FTHS). This study extends previous analyses of this mouse model, providing an in-depth characterization of the calvarial phenotype. The study uses a combination of microCT imaging, histology and in vitro assays to demonstrate reduced osteogenesis in the Sh3Pxd2b (nee) mouse model. The study further uses in vitro assays to show reduced cell migration and reduced proliferation in the Sh3Pxd2b-/- mice. There are significant issues with the image quality for some of these studies, making it difficult to interpret the data. The last part of the study uses RNA-seq to explore potential molecular mechanisms, and explores downregulation of ribosome biogenesis; however, I have significant concerns regarding the interpretation of this data, which is detailed below. Overall, this study provides a more comprehensive picture of the Sh3Pxd2b-/- phenotype and has potential to provide mechanistic information if additional studies are performed.

SUGGESTIONS TO AUTHORS

1. The skull is represented in multiple images in Figure 1 using microCT (B), alizarin red (C), and calcein staining (D), and histology (E). These images appear to have been cropped so that portions of the skull are removed from the images, making it difficult to interpret and compare the

images. The zygomatic is missing from the microCT images in both control and mutant in panel B. The nasal bone is largely missing from all images in panels C and D in addition to portions of the calvarial bones. Including the full images would improve the quality of the figure as well as making clear the interpretation of the data. 2. The images in Figure 1 are very small and difficult to interpret. Please consider moving some of this data to supplementary figures so that the images can be presented clearly.

Author's reply: We thank the reviewer for the suggestions. The authors want to clarify, that it is not the images that have been cropped, but the skull itself to focus on the areas of interest, being mainly the PF and the Sag suture. The images have then been placed on a white background. To address the reviewers' concern, we reconstructed the Micro-CT images in Panel B to show the zygomatic bone. Additionally, the authors added Panel C to show Alizarin Red whole mount staining of the whole skull, including the nasal bone to allow for comparisons between Panels B and C.

To assess the evolution of the non-mineralized areas along the midline of the skull, Panels D and E show Alizarin Red and Calcein stained images of the area of interest from the lamboid suture to the Bregma at different time points. The images have been adapted to only show the areas of interest over time. This also created more space so that Panels A-C could be increased in size for better visibility. The text was adapted in the second paragraph of the Results Sections (line 92-98) and in the Figure 1 legend to incorporate the changes.

3. The figure legends state experiments were performed in triplicate with n=3 per group, but it is not clear whether these were littermates, and whether littermates were sex matched. Please more clearly indicate this information.

Author's reply: Littermates of the same sex (male and female) were used for experiments. This is described in the section "Animal husbandry" of Materials and Methods.

4. In Figure 2, since the wt mice do not show much bone regeneration, this could suggest increased osteoclast activity. Is there any evidence for increased osteoclast activity, for example TRAP staining? And if there is an imbalance in bone homeostasis, does this contribute to the suture patency over time? Could the suture phenotypes be a combination of both reduced osteogenesis and increased osteoclast activity?

Author's reply: We thank the reviewer for this important comment. In response, we performed additional TRAP staining analyses to examine osteoclast activity in both the posterior frontal (PF) suture and calvarial defect models of *Sh3Pxd2b*^{nee -/-} and WT mice.

In the PF suture, TRAP-positive osteoclasts were present along the suture margins and within resorption pits in WT mice, consistent with active bone remodeling during physiological suture fusion. In contrast, mutant mice showed delayed and spatially restricted osteoclast activity, with TRAP-positive cells retracted to the lateral bone fronts and absent from the central suture mesenchyme, suggesting impaired osteoclast localization and remodeling capacity.

In the calvarial defect model, both WT and mutant mice exhibited TRAP-positive cells at the bone fronts, but no TRAP activity was detected within the defect center in either WT or mutant mice. Notably, the defect area in *Sh3Pxd2b*^{nee -/-} mice progressively enlarged over time, indicating that impaired bone regeneration was not due to excessive osteoclast activity, but rather reflects deficient osteogenic repair.

We have revised the Results and Discussion section accordingly to reflect these findings (lines 142-155, 161-171, 396-397, 404-411).

5. The localization of the antibody stain in Figure 3 would benefit from additional explanation. Given the *nee* mutation leads to a truncated protein, this seems related to the work of Adam et al., 2015 that showing changes in localization of Tks4 truncated protein, as well as localization to aggresomes, which would be worth citing. The images in the osteoblasts do not seem to show any differences between control and mutant cells with respect to the Tks4 localization; and it

seems that there are more perinuclear condensates in wt versus mutants in the images in the figure. Given that the conclusion was mutants were more condensed, is there quantification to demonstrate this? What percentage of cells shows the localization pattern represented?

Author's reply: We thank the reviewer for this comment regarding TKS4 localization. In 44 counted mutant dura mater cells and 35 counted mutant osteoblasts, we observed a perinuclear condensation of TKS4 in 75% and 57% of cells, respectively. The altered intracellular TKS4 distribution is more pronounced in dura mater cells than in osteoblasts. Please also refer to Figure 7, showing perinuclear accumulation with focal condensation points in mutant osteoblasts. The quantification has been added in the Results section from lines 187-190. Additionally, we also address the reviewer's suggestion to contextualize these findings by citing Ádám et al. (2015), who reported aberrant intracellular distribution and aggresomal accumulation of mutated Tks4 proteins. While we did not directly assess aggresome formation, the similarities in altered TKS4 localization we observe may reflect a similar phenomenon. We added a brief discussion of the cited paper in line 361-371.

6. All sections from E9.5 embryos need anatomical landmarks (Figs 3, 4, 5 and 7) to show the sections are from the same location in the embryo. It is unclear which specific region "head area" refers to in Figure 3. Showing the entire section with anatomical landmarks would be beneficial to orient readers to where in embryo the expression pattern was examined.

Author's reply: We appreciate the reviewer's concern. While corresponding whole sections of the embryos are not available, we have included a whole-mount image of a representative wild-type E9.5 mouse embryo to support anatomical orientation. This image is presented in Figure S2A and respective references have been made in the text when describing Figure 3C, 4D, 5A-D, 7E-F. Key anatomical landmarks have been annotated. The presumptive head region and neural plate (NP) areas used for subsequent immunofluorescent imaging have been clearly marked on the embryo, allowing spatial context and reproducibility of regional selections. We believe this addition clarifies the anatomical basis for our region-specific assessments.

7. The DAPI in Fig. 3 panel C seems to show different orientation and the nuclei appear larger in the mutant image- were these images taken at the same magnification? The staining for Tks4 also does not seem to be very specific in these sections - is it possible that there was just high background? At E9.5, the staining looks reduced in wt compared to mutant, in contrast to the statement of no difference.

Author's reply: We thank the reviewer for the helpful observations. We have replaced the WT E9.5 image in Fig. 3 panel C with a new representative section in which both the DAPI and Tks4 staining are clearly in focus and at the same magnification for wild-type and mutant embryos. The previous image likely appeared inconsistent due to a minor focal plane difference, which may also have contributed to the perception of increased background in the Tks4 staining. With the updated image, the nuclear morphology is consistent between WT and mutant mice, and the Tks4 signal more clearly reflects the described expression pattern.

8. Is the noted difference in Tks4 expression in Fig. 3D a true reduction in the mutant, or just a lack of those cell populations? The differences stated in the text about Fig. 3 are not easily distinguished and could benefit from image analysis to support the conclusions.

Author's reply: We thank the reviewer for this important observation. The noted reduction in Tks4 staining in Figure 3D may very well reflect the disruption of tissue architecture in the PF suture of *Sh3Pxd2b^{hee} -/-* mice. In the mutant, the suture mesenchyme, periosteum, and dura mater appear disorganized or partially absent, which may contribute to the observed difference in TKS4 staining. We agree that distinguishing between reduced expression and absence of specific cell populations is important, and we have clarified this point in the revised text (lines 195-197). While we have not performed cell-type-specific quantification, the altered anatomical structure in the mutant PF suture strongly suggests that the difference may be due in part to the absence of the relevant cell populations. This clarification has been added to the Results section to better reflect the complexity of the interpretation.

9. The EdU proliferation assay shows a change in vitro, but additional quantification would be beneficial on the sections from E9.5 and pN3. In Fig. 4D, what do "outer tissue layers" refer to? The morphology of the tissue sections between control and mutant look different here, so are differences in signal due to a different area of the embryo? Including the entire section for anatomical reference would be beneficial.

Author's reply: We appreciate the reviewer's suggestion regarding quantification and clarification of the anatomical context in Fig. 4D. In this panel, the term "outer tissue layers" refers to the surface ectoderm and adjacent mesenchymal layers of the presumptive craniofacial region in the E9.5 embryo and the text has been adjusted accordingly. We agree that overall tissue morphology differs between WT and mutant embryos, which likely reflects underlying developmental defects and altered patterning in the mutants. To aid in anatomical orientation, we have now included a representative image of a whole-mount WT E9.5 embryo with labeled regions of interest in Supplemental Figure 2A.

Regarding quantification, we have performed and included quantification for the in vitro EdU proliferation assays, where individual cell counts and morphologies are well-defined and directly comparable. However, for the in vivo E9.5 and pN3 tissue sections, our aim was primarily to highlight the distinct spatial distribution of proliferative cells rather than to perform absolute quantification, which is complicated by differences in tissue architecture and cell density. We believe that the qualitative differences in proliferative localization are biologically meaningful and informative in the context of the observed craniofacial phenotype.

10. In Fig. 4E and F, is the noted increase in proliferation in the mesenchyme in wt simply because there are more cells in wts? Quantification of the percentage of PCNA+ cells/DAPI would be informative here since the morphology between controls and mutants is already quite different.

Author's reply: We thank the reviewer for this thoughtful comment. While we have performed quantification of proliferative differences using EdU incorporation in vitro, quantifying PCNA+ cells in the tissue sections shown in Figures 4D-F is challenging due to the different tissue architecture between WT and mutant mice. Specifically, the organization and cellularity of the suture region vary significantly, making direct comparison based on total DAPI+ cells potentially misleading. However, we believe that the different spatial distribution of proliferative cells reflects a biologically meaningful difference. For instance, the mutant suture mesenchyme is devoid of PCNA-positive cells, which may be a result of defective recruitment or maintenance of cephalic neural crest-derived progenitor populations that normally populate this region. We have added some additional clarifying text to emphasize the spatially restricted localization of proliferative cells in the Results and the Discussion from lines 208-2011, 259, 379, 386-388.

11. TGF β treatment did not appear to increase podosome formation in controls (data in Fig. 4I). Is there an explanation for this?

Author's reply: Thank you for the comment. While TGF β is a known promoter of cell migration and has been shown to induce podosome formation in endothelial cells and fibroblasts, its role is complex and not fully understood across all cell types. We would like to emphasize that this experiment was intended as a supportive rather than conclusive test of our hypothesis using a known effector of migration. We observed a trend towards increased podosome formation in dura mater cells and osteoblasts with TGF β treatment, but this increase was not statistically significant, which may reflect cell type-specific differences or additional cofactors or specific conditions to fully activate podosome formation in these cells. Further investigation is needed to better understand the nuanced effects of TGF β on podosome dynamics in these cell types, however we feel this is out of the scope of this manuscript.

12. The images in Fig. 4M, N need to represent the full section. As the data is currently represented, it is not possible to determine if there is a reduction in migration or if a portion of the image is simply missing. The conclusion that migration is more affected in cells of neural crest cell origin requires additional support or explanation. It is not clear from the materials and

methods that the osteoblasts are solely those of neural crest cell origin, or whether they are a mixed population of neural crest and mesoderm derived, or only mesoderm. If mesoderm versus neural crest derived osteoblasts were examined, please include that information. The images in Fig. 4 M, N are not clear enough to fully interpret, but it would appear that neither mesoderm nor neural crest derived cells are migrating.

Author's reply: We appreciate the reviewer's feedback and agree that clarification regarding the origin of the migrating cells and the completeness of the sections is important. To address this concern, we have provided additional representative images of the full calvarial defect area for Figure 4M in new Supplemental Figure S3, which illustrate the overall distribution of migrating cells in WT defects, and the lack thereof in defects of mutant mice. For Figure 4N, the fluorescent nature of the sample limited our ability to re-image the entire section. However, the region shown is representative of the defect area, and the imaging field was chosen to reflect comparable positions in the wild-type and mutant mice.

In the *in vivo* calvarial defect assay shown in Figure 4M-N, the defect was intentionally created in the parietal bone, which is of mesodermal origin and therefore not expected to include neural crest-derived osteoblasts. Dura mater cells, which originate from a thin layer of neural crest cells, have been shown to play a key role in calvarial defect repair (Levi et al., 2011, Greenwald et al., 2000, Spector et al., 2002). To assess whether the migrating cells were of neural crest origin, we used a *Wnt1-Cre2* transgenic mouse expressing Cre recombinase under the control of the neural crest marker *Wnt1*, with a double fluorescent *mT/mG* reporter mouse, used for lineage tracing. The cells migrating into the defect area in the WT mouse express GFP and are thus of neural crest origin, likely originating from the dura mater. We cannot fully exclude the presence of GFP-negative migratory cells in the defect area of WT mouse, including mesoderm derived osteoblasts from the surrounding parietal bone. However, though we did not distinguish between osteoblasts of mesodermal and neural crest origin in the *in vitro* migration assays, we did not see any migratory impairment in osteoblasts *in vitro* (Figure 4G). In contrast, migration was significantly decreased in neural crest-derived dura mater cells *in vitro* (Figure 4G and H). In addition, we did not observe any cell migration into the calvarial defect area in mutant mouse, while GFP-expressing cells of neural crest origin did migrate into the defect area of WT mice. Taken together, this data is suggestive of migratory impairment in the neural crest-derived cell population.

13. Fig. 5C is a repeat of what was already in Fig. 4D, and Fig 5B is a repeat of Fig. 3C. It seems more logical to have this data just once, and having this younger data altogether in Fig. 5 makes sense. Please provide anatomical information for the sections.

Author's reply: We thank the reviewer for this observation. While Figures 5B and 5C display staining for the same markers as in Figures 3C and 4D, respectively, they are derived from distinct anatomical regions and serve a different purpose within the study. Specifically, Figures 5B and 5C show immunofluorescent staining in the neural plate (at the level of the midbrain-hindbrain boundary) and are intended to highlight defects in early neural crest progenitor populations. In contrast, Figures 3C and 4D depict staining from the presumptive head region of E9.5 embryos.

A schematic overview of an E9.5 embryo is depicted in Figure 5 with a window indicating the location from which each section (5A-5D) was taken. Furthermore, we now include a representative whole-mount image of an E9.5 embryo with anatomical landmarks and region-specific section windows in Supplemental Figure S2A, providing better anatomical context for these images.

14. The TUNEL staining looks increased in the mutants in Fig. 5D - was this true across multiple sections from multiple embryos? The text states that TUNEL is not changed.

Author's reply: We appreciate the reviewer's careful observation. While the fluorescence in Figure 5D may appear increased, we believe this signal represents background fluorescence rather than true apoptotic staining. We have included positive controls for the TUNEL assay using DNase-treated samples, as recommended by the manufacturer, in Supplemental Figure S2B and C for *in vitro* assays and S2D and E for *in vivo* sections. These demonstrate the expected distinct nuclear

staining pattern for apoptotic cells, which is not observed in Figure 5D, supporting our conclusion that apoptosis is not significantly altered in the mutant samples.

15. The explant migration assays are informative for examining podosomes, but is there evidence of a neural crest cell migration defect *in vivo*? Does the overall migration pattern of neural crest cells appear abnormal at E9.5 or E10.5 in the mutant embryos with Wnt1Cre;mT/mG? When do differences in the craniofacial morphology appear in the *Sh3Pxd2b*^{-/-} mice? This is an important point regarding mechanism. If there are already changes in proliferation and migration at E9.5 that lead to the differences observed postnatally, then this would be important to examine in greater detail.

Author's reply: We appreciate the reviewer's important point regarding *in vivo* neural crest cell migration. We observed decreased cell proliferation in sections of the mutant embryonic neural plate *in vivo* and impaired migration in mutant embryonic neural plate explant assays and show *in vivo* data suggestive of impaired neural crest-derived cell migration in a postnatal calvarial defect model. Together, these findings are indicative of an abnormal neural crest cell proliferation and migration pattern established during embryogenesis and retained postnatally. We acknowledge that direct live-imaging *in vivo* evidence of early neural crest cell migration abnormalities in *Sh3Pxd2b*^{nee -/-} embryos at E9.5 or E10.5 would be important to examine in greater detail. However, real-time *in vivo* imaging and high-resolution lineage tracing at these early embryonic stages require a specialized embryology setup, including live imaging systems and advanced microsurgical techniques, which are currently not available in our laboratory.

16. The RNA-seq experiment was performed at pN15 on whole skulls (mesoderm and neural crest origin combined). This time point is well after defects in the mutant mice are apparent, better reflecting the role for *Sh2Pxd2b* during postnatal growth. Since the previous section seemed to be suggesting that *Sh3Pxd2b* had an early role in neural crest cell development, could you provide some rationale for choosing pN15?

Author's reply: We thank the reviewer for this comment. We chose postnatal day 15 (pN15) for RNA-seq analysis as it represents a stage of active skull growth and remodeling. At this time point, critical processes such as osteoblast differentiation, matrix deposition, and suture remodeling are still ongoing in both neural crest- and mesoderm-derived bones. Performing transcriptomic profiling at this stage allows us to capture gene expression changes that may contribute to or result from the established craniofacial abnormalities observed in *Sh3Pxd2b*^{nee -/-} mice. Our goal was to provide a global view of molecular alterations associated with the postnatal craniofacial phenotype. We agree that earlier embryonic or early postnatal time points could yield important insights into the initial molecular events leading to the observed phenotype. These will be valuable in future studies aimed at understanding the developmental timing and primary mechanisms underlying the role of *Tks4* in craniofacial development.

17. Please include more information about the RNA-seq experiment in the materials and methods section with details on the library preparation, quality control cutoffs, and analyses. For example, was there a minimum expression threshold used (ie TPM>2)? Was p value or adjusted p value used for cutoff (text states p and not adjusted p, which should be used to account for multiple testing)? If the library was made using polyA selection, differences in ribosomal RNA and other non-coding RNAs should be interpreted with caution and carefully validated. If a significant difference is seen in 5S rRNA in the mutant versus control sample using an assay like the Agilent Bioanalyzer or Northern blot, then this would support a true change versus an artifact from the library preparation method. For the KEGG analysis, please include the table on the effect size and direction - are all of the genes presented in Fig. 7E downregulated, or were some upregulated?

Author's reply: We appreciate the reviewer's comment and have now expanded the Materials and Methods section accordingly (lines 628-638).

In brief, following alignment to the *mm10* reference genome using HISAT and transcript assembly in StringTie with default settings (e.g., minimum isoform fraction >0.15, minimum assembled transcript length >200), the resulting sample-level feature matrices were processed using the

standard EdgeR pipeline with normalization by trimmed mean of M values (TMM). Genewise exact tests were applied via built-in EdgeR functions to identify differentially expressed genes between mutant and wild-type groups, using a false discovery rate (FDR)-adjusted p-value threshold of <0.05 for downstream analysis. We did not use a minimum expression cutoff such as TPM >2; however, the EdgeR pipeline includes statistical modeling of low counts and robust dispersion estimates to control for low-expression noise.

The RNA libraries were prepared using a polyA enrichment method, which leads to depletion of non-polyadenylated RNAs, including ribosomal RNAs and small nucleolar RNAs. Therefore, we agree, that transcript-level findings involving 5S rRNA and other non-coding RNAs should be interpreted with caution and we thank the reviewer for bringing this to our attention.

Of note, the same library preparation protocol was applied consistently across all WT and mutant samples, thus the observed reduction in 5S rRNA might reflect a true biological difference rather than a technical artifact. To further validate this, we performed additional analysis using the Agilent Bioanalyzer Small RNA assay as suggested by the reviewer. This independent analysis revealed a consistent reduction in the 5S rRNA peak in mutant samples, as well as decreased overall small RNA concentration and reduced miRNA proportion, reinforcing the transcriptomic findings and suggesting a broader alteration in small RNA homeostasis. These results are included in the revised Results (line 309-313) and shown in Figure 6E and Supplemental Table 2.

Regarding the KEGG pathway enrichment analysis shown in Figure 7E, we confirm that all genes used were downregulated in the mutant samples. We have added a supplementary table with effect sizes in Supplemental Table 1, as requested.

We hope these clarifications and additions adequately address the reviewer's concerns and strengthen the interpretation of our data.

18. The upregulated genes mentioned that were metabolic and biochemical could still be of relevance in this system, especially given potential changes in bone homeostasis (Fig 2). This point could be worth further consideration in the discussion.

Authors reply: We appreciate the reviewer's thoughtful comment regarding the potential relevance of upregulated genes involved in metabolic and biochemical pathways, particularly in the context of altered bone homeostasis. While many of these genes appear unrelated to craniofacial development at first glance, we agree that these pathways may be relevant in the broader context of bone homeostasis, potentially through indirect effects. However, to maintain focus on the primary mechanisms explored in this study, we have opted not to expand the discussion of these genes at this time.

19. The Y10b antibody staining seems to suggest that rRNA is not downregulated in mutants relative to controls, in contrast to the RNA-seq result. Can you explain this? (However, it should be noted that Y10b is not a marker of 5S rRNA specifically, just rRNA more broadly.)

Author's reply: We thank the reviewer for this important point. Indeed, we observed that Y10b immunoreactivity was not globally reduced in mutant tissues, while the RNA-seq data indicates downregulation of rRNAs, including nuclear-encoded 5S rRNA and small nucleolar RNAs involved in ribosome biogenesis.

Y10b is a monoclonal antibody that broadly detects ribosomal RNA. Most rRNA in the cytoplasm is part of fully assembled ribosomes or ribosomal subunits and thus, Y10b staining generally reflects the presence and localization of mature or assembled rRNA within the cell, rather than the transcriptional output or availability of specific rRNA precursors (such as 5S rRNA or pre-rRNA intermediates).

In contrast, RNA-seq on postnatal calvarial bone provides a bulk, transcription-level snapshot that includes downregulation of ribosome biogenesis pathways and precursors, including transcripts like 5S rRNA (which is nuclear-encoded) and multiple snoRNAs. These RNAs are critical for ribosome assembly but are not directly targeted by Y10b. Therefore, the two methods reflect different

biological layers:

- RNA-seq shows reduced transcription of rRNA-related genes, possibly indicating impaired ribosome production capacity.
- Y10b staining reflects existing pools of mature rRNA and ribosomes, which may persist or accumulate aberrantly in specific cellular compartments (e.g., cell protrusions), as we observe in NCCs of *Sh3Pxd2b^{nee -/-}* mice.

20. There is no experimental evidence provided to support downregulation of ribosome biogenesis. Including experiments to show downregulation of ribosome biogenesis in the *Sh3Pxd2b* mutant embryos would provide preliminary support for the model proposed in Fig. 7G.

Author's reply: We completely agree with the reviewer that direct experimental evidence supporting a defect in ribosome biogenesis is needed to strengthen our hypothesis. In response, we have now included the additional data obtained using the Agilent Bioanalyzer Small RNA assay, as described in the response to comment #17. While our findings are suggestive of altered small RNA profiles and may reflect disruptions in ribosome-related processes, we recognize that they do not provide direct mechanistic evidence of impaired ribosome biogenesis.

To reflect this, we have revised the relevant text in the abstract, results and discussion sections (line 309-313, 349-355, 428-449) to soften our conclusions and clearly state that further studies are needed to confirm any disruption in ribosome biogenesis. Additionally, to avoid overinterpretation, we have removed the schematic model previously shown in Figure 7G. We believe these changes more accurately reflect the preliminary nature of the findings while still highlighting potential areas of future investigation.

21. The colocalization of Y10b and Tks4 is perhaps not surprising given both are broadly expressed. This is correlative, and further evidence is necessary to determine if this is meaningful, which should be highlighted in the discussion. The suggestion of FTHS as a ribosomopathy should be avoided until there is definitive experimental evidence supporting diminished ribosome biogenesis.

Author's reply: We thank the reviewer for raising this important point. We fully agree that the immunofluorescence-based analysis used in our study demonstrates spatial signal overlap between Y10b and TKS4, rather than definitive molecular colocalization. To address this, we have revised the text in the results and discussion sections to clarify that these findings are correlative. Specifically, we now refer to "overlap" rather than true colocalization (lines 339-343, 354-355, 439-440, Figure 7 legend).

In addition, we acknowledge that our data do not provide sufficient experimental evidence to support classification of FTHS as a ribosomopathy. We have removed this term from the manuscript and adjusted our conclusions accordingly. While transcriptomic data, altered rRNA localization, and reduced 5S rRNA levels (supported by new Bioanalyzer analysis described in comment #17) suggest potential involvement of ribosome-related processes, we clearly state that these findings are preliminary and that further studies are needed to assess ribosome biogenesis and function in mutant cells.

22. There is a point in the discussion about migration defects specifically in neural crest cell derived tissues, but it seems likely that there are also these migration defects in mesoderm-derived calvarial bones based on the data presented in Fig. 4.

Author's reply: We thank the reviewer for this insightful comment. In our *in vivo* calvarial defect model (Figure 4M-N), the defect was created in the parietal bone, which is of mesodermal origin and not expected to include neural crest-derived osteoblasts. Repair of the calvarial defect is likely mediated by underlying dura mater, which arises from a thin layer of neural crest-derived cells. Using *Wnt1-Cre2; mT/mG* reporter mice, we observed GFP-positive cells—indicative of

neural crest origin—migrating into the defect in WT mice but not in *Sh3Pxd2b^{nee -/-}* mice. This supports a migratory defect in the neural crest-derived population.

In our *in vitro* assays, we did not distinguish between osteoblasts of neural crest (frontal bone) and mesodermal (parietal bone) origin, resulting in a mixed osteoblast population. However, no migration defect was observed in mutant osteoblasts *in vitro* (Figure 4G). In contrast, migration of dura mater cells—known to be neural crest-derived—was significantly impaired *in vitro* in *Sh3Pxd2b^{nee -/-}* mice (Figure 4G,H). This differential outcome further suggests that the migration defect is more prominent in neural crest-derived populations.

While we cannot completely exclude subtle or context-specific effects on mesoderm-derived cells, our *in vivo* and *in vitro* data together point to a primary impairment in neural crest-derived cell migration.

Minor points

- There is aberrant text over a couple of panels in Figure 1 part F; and "merged" is cut off on the bottom portion of the text

Author's reply: We thank the reviewer for noticing and have revised the Figure accordingly.

- Mouse nomenclature for protein is Tks4 (only the first letter capitalized)

Author's reply: We thank the reviewer for the comment. According to Mouse Genome Informatics (MGI), protein symbols in mice should use all uppercase letters and are not italicized.

<https://www.informatics.jax.org/mgihome/nomen/gene.shtml#ps>

- Spell out EdU the first time and then use the abbreviation.'

Author's reply: We thank the reviewer for the comment and have revised the text accordingly.

References:

- BÖGEL, G., GUJDÁR, A., GEISZT, M., LÁNYI, Á., FEKETE, A., SIPEKI, S., DOWNWARD, J. & BUDAY, L. 2012. Frank-ter Haar syndrome protein Tks4 regulates epidermal growth factor-dependent cell migration. *J Biol Chem*, 287, 31321-9.
- BRADLEY, J. P., LEVINE, J. P., MCCARTHY, J. G. & LONGAKER, M. T. 1997. Studies in cranial suture biology: regional dura mater determines *in vitro* cranial suture fusion. *Plast Reconstr Surg*, 100, 1091-9; discussion; 1100-2.
- BUSCHMAN, M. D., BROMANN, P. A., CEJUDO-MARTIN, P., WEN, F., PASS, I. & COURTNEIDGE, S. A. 2009. The novel adaptor protein Tks4 (SH3PXD2B) is required for functional podosome formation. *Mol Biol Cell*, 20, 1302-11.
- GREENWALD, J. A., MEHRARA, B. J., SPECTOR, J. A., CHIN, G. S., STEINBRECH, D. S., SAADEH, P. B., LUCHS, J. S., PACCIONE, M. F., GITTES, G. K. & LONGAKER, M. T. 2000. Biomolecular mechanisms of calvarial bone induction: immature versus mature dura mater. *Plast Reconstr Surg*, 105, 1382-92.
- HOBAR, P. C., SCHREIBER, J. S., MCCARTHY, J. G. & THOMAS, P. A. 1993. The role of the dura in cranial bone regeneration in the immature animal. *Plast Reconstr Surg*, 92, 405-10.
- JIANG, X., ISEKI, S., MAXSON, R. E., SUCOV, H. M. & MORRIS-KAY, G. M. 2002. Tissue origins and interactions in the mammalian skull vault. *Dev Biol*, 241, 106-16.
- KUDLIK, G., TAKÁCS, T., RADNAI, L., KURILLA, A., SZEDER, B., KOPRIVANACZ, K., MERŐ, B. L., BUDAY, L. & VAS, V. 2020. Advances in Understanding TKS4 and TKS5: Molecular Scaffolds Regulating Cellular Processes from Podosome and Invadopodium Formation to Differentiation and Tissue Homeostasis. *Int J Mol Sci*, 21.
- LEVI, B., NELSON, E. R., LI, S., JAMES, A. W., HYUN, J. S., MONTORO, D. T., LEE, M., GLOTZBACH, J. P., COMMONS, G. W. & LONGAKER, M. T. 2011. Dura mater

- stimulates human adipose-derived stromal cells to undergo bone formation in mouse calvarial defects. *Stem Cells*, 29, 1241-55.
- MAO, M., THEDENS, D. R., CHANG, B., HARRIS, B. S., ZHENG, Q. Y., JOHNSON, K. R., DONAHUE, L. R. & ANDERSON, M. G. 2009. The podosomal-adaptor protein SH3PXD2B is essential for normal postnatal development. *Mamm Genome*, 20, 462-75.
- MEHRARA, B. J., GREENWALD, J., CHIN, G. S., DUDZIAK, M., SAGRIOGLU, J., STEINBRECH, D. S., SAADEH, P. B., GITTES, G. K. & LONGAKER, M. T. 1999. Regional differentiation of rat cranial suture-derived dural cells is dependent on association with fusing and patent cranial sutures. *Plast Reconstr Surg*, 104, 1003-13.
- MOSSAZ, C. F. & KOKICH, V. G. 1981. Redevelopment of the calvaria after partial craniectomy in growing rabbits: the effect of altering dural continuity. *Acta Anat (Basel)*, 109, 321-31.
- SPECTOR, J. A., GREENWALD, J. A., WARREN, S. M., BOULETREAU, P. J., CRISERA, F. E., MEHRARA, B. J. & LONGAKER, M. T. 2002. Co-culture of osteoblasts with immature dural cells causes an increased rate and degree of osteoblast differentiation. *Plast Reconstr Surg*, 109, 631-42; discussion 643-4.
- YU, J. C., MCCLINTOCK, J. S., GANNON, F., GAO, X. X., MOBASSER, J. P. & SHARAWY, M. 1997. Regional differences of dura osteoinduction: squamous dura induces osteogenesis, sutural dura induces chondrogenesis and osteogenesis. *Plast Reconstr Surg*, 100, 23-31.

Second decision letter

MS ID#: dev.204631R1

MS TITLE: The Sh3Pxd2bnee^{-/-} mouse: an attractive model unveiling novel developmental features in Frank-ter-Haar Syndrome

AUTHORS: Julika Huber, Siddharth Menon, Michael Lopez-Torres, Jason L. Guo and Michael T. Longaker

Dear Dr Huber,

Thank you for your cooperation in sharing the original data for your manuscript. We requested this information because one of the reviewers suggested that the figures were altered without being detailed in the materials and methods. In any case, all figures in accepted papers are screened as part of our standard integrity checks before publication, so any issues would have been raised then. The referees' comments are appended below, or you can access them online: please go to:

As you know, we've investigated this case and you can find the full report attached to this letter. Although we do not suggest any deliberate intention to mislead, we have concluded that image adjustments are excessive, do not adhere to best practice and could affect the interpretation of the data. Therefore, it is essential that you remake the figures (following the guidance detailed below) and that the revised figures are seen by the reviewers to ensure their interpretation has not changed and the conclusions of the paper remain supported by the data.

In your revised figures, please ensure that:

- any adjustments, edits or filters used in processing micrograph images are applied to the entire image and specific parts of individual images are not selectively edited.
- the same adjustments (if any) should be applied to corresponding images where direct comparisons are drawn (e.g. control vs experimental data)
- it is clearly annotated or indicated when figure panels are composed, stitched or tiled from two or more individual images.
- the original aspect ratio is maintained when resizing images to prevent stretching or skewing.
- cropped images remain representative of the object of interest.
- adjustments or processing of the images are discussed/explained in detail in the materials and methods.

If you are able to revise the manuscript along the lines suggested, which may involve further experiments, I will be happy to receive a revised version of the manuscript. Your revised paper will be re-reviewed by one or more of the original referees, and acceptance of your manuscript will

depend on your addressing satisfactorily the reviewers' major concerns. Please also note that Development will normally permit only one round of major revision. If it would be helpful, you are welcome to contact us to discuss your revision in greater detail. Please send us a point-by-point response indicating your plans for addressing the referees' comments, and we will look over this and provide further guidance.

Please attend to all of the reviewers' comments and ensure that you upload both a 'clean' version of your Word file, along with a highlighted version clearly showing where you have made changes in the revised manuscript. Please avoid using 'Tracked changes' in Word files as these are lost in PDF conversion. I should be grateful if you would also provide a point-by-point response detailing how you have dealt with the points raised by the reviewers in the 'Response to Reviewers' box. If you do not agree with any of their criticisms or suggestions please explain clearly why this is so.

Reviewer 1

Advance summary and potential significance to field

The reviewers have addressed all my questions and also clarified the nature of the (truncating) mutation.

I have no further questions and only one very minor correction:

Lines 141-143: The sentence "we observed increased bone nodules formation and matrix mineralization in WT osteoblasts compared to mutant osteoblasts as assessed by Alizarin Red staining and quantification (Figure 2A,B)" should be revised. The authors could state that bone nodule formation is higher in WT than in mutant, but it is not correct correct to say that this process is "increased" in WT.

Reviewer 2

Advance summary and potential significance to field

In this revised manuscript from Huber et al, a detailed description of postnatal craniofacial phenotypes in a mouse model of Frank-ter-Harr Syndrome (FTHS) is presented. The studies utilize the Sh3Pdx2bnee/nee mutant mouse model, in which a truncated protein is expressed. Sh3Pdx2bnee/nee mutant mice show changes in the development of the skull and sutures, and compelling evidence is provided for impaired osteogenic capacity of mutant cells during postnatal development both in the PF suture and upon induction of a calvarial defect. To further assess the underlying mechanisms, the expression of TSK4, proliferation, cell death, and migration were investigated both in vitro and in vivo. Podosome formation was reduced in both dura mater cells and osteoblasts, but in vitro assays of migration indicated that only mutant dura mater cells display reduced migration. The cranial neural plate explant studies suggest reduced migration and altered podosome structure in neural crest cells, which give rise to dura mater cells. Together, this suggests that migration deficiencies are not the same across all tissues and is an interesting point in the study. RNA-seq was performed at P15 to further interrogate mechanisms of skull growth, which revealed that most genes were upregulated, with fewer genes being downregulated and these were associated with diminished ribosome biogenesis. The authors note a difference in the 5S rRNA, but this remains to be quantified (see note below). Altogether, these studies provide new insight into the mechanism of differences in skull development in a mouse model of FTHS, extending previous studies and providing a foundation for multiple areas of future research.

Comments for the author

1. The authors have carefully responded to reviewer comments and softened the language surrounding ribosome biogenesis to prevent its over-interpretation. However, the analysis of small RNAs requires further consideration in both how the data was analyzed and its interpretation. Based on the data presented, the suggestion of disrupted ribosome biogenesis (line 320) is still not supported. For comparison of 5S rRNA, the exact same amount of total RNA should be loaded for

each sample, which is not noted in the materials and methods. If this is indeed the case, please note this in the methods. To assess specific downregulation of 5S rRNA, the peaks should be quantified as a ratio (see Bir et al., 2024; PMID: 38192358 for an example of small RNA interpretation). As the data is currently represented in Fig. 6E, it appears that the mutant samples have less overall RNA as the pattern of the peaks does not appear to be significantly different across samples and all peaks are reduced in mutant samples. To be able to conclude a reduction in the 5S rRNA, this should be carefully quantified and statements in the discussion (lines 439-441) adjusted accordingly.

2. Y10b localizes to protrusions in WT but not mutant dura mater cells, and then increased Y10b is observed in protrusions on mutant cultured NCCs. This seems to suggest that what might be happening in terms of Y10b and migration is different in NCCs versus dura mater cells, which could be an interesting area for future investigation.

Minor points

1. The description of mutant osteoblasts treated with TGFB is missing from the section in lines 250-253.
2. Suggest rephrasing the statement regarding Fic 3C (line 197) to say that the staining in the PF suture is decreased in mutants as opposed to increased in WT
3. Fig. 1 legend (line 892) missing "6" when referring to the postnatal month
4. The image in S2A is helpful for addressing the areas imaged in Fig. 3C and Fig 4D and Fig. 7. To further improve the clarity, including a labeled dashed line to indicate plane of section for 3C and 4D (as well as 7E and 7F) would also be helpful. The sectioning of embryos should also be noted in the materials and methods section which is currently missing information on embryonic tissue harvesting and histology.
5. Similarly, indicating the plane of section in figure 5 would also be helpful.
6. Capitalize S in 5S line 444
7. In the author contributions, the initials N.Q. are listed and do not appear to correspond with the listed authors.

Second revision

Author response to reviewers' comments

Reviewer 1: The reviewers have addressed all my questions and also clarified the nature of the (truncating) mutation.

I have no further questions and only one very minor correction:

Lines 141-143: The sentence "we observed increased bone nodules formation and matrix mineralization in WT osteoblasts compared to mutant osteoblasts as assessed by Alizarin Red staining and quantification (Figure 2A,B)" should be revised. The authors could state that bone nodule formation is higher in WT than in mutant, but it is not correct correct to say that this process is "increased" in WT.

Author's reply: We agree with the reviewer's observation and have changed the wording to the following: "We observed decreased bone nodules formation and matrix mineralization in mutant osteoblasts compared to WT osteoblasts as assessed by Alizarin Red staining and quantification" now in line 131-132.

Reviewer 2: SUMMARY OF THE ADVANCE MADE IN THIS PAPER AND ITS POTENTIAL SIGNIFICANCE TO THE FIELD

In this revised manuscript from Huber et al, a detailed description of postnatal craniofacial phenotypes in a mouse model of Frank-ter-Harr Syndrome (FTHS) is presented. The studies utilize the Sh3Pdx2bnee/nee mutant mouse model, in which a truncated protein is expressed. Sh3Pdx2bnee/nee mutant mice show changes in the development of the skull and sutures, and compelling evidence is provided for impaired osteogenic capacity of mutant cells during postnatal

development both in the PF suture and upon induction of a calvarial defect. To further assess the underlying mechanisms, the expression of TSK4, proliferation, cell death, and migration were investigated both in vitro and in vivo. Podosome formation was reduced in both dura mater cells and osteoblasts, but in vitro assays of migration indicated that only mutant dura mater cells display reduced migration. The cranial neural plate explant studies suggest reduced migration and altered podosome structure in neural crest cells, which give rise to dura mater cells. Together, this suggests that migration deficiencies are not the same across all tissues and is an interesting point in the study. RNA-seq was performed at P15 to further interrogate mechanisms of skull growth, which revealed that most genes were upregulated, with fewer genes being downregulated and these were associated with diminished ribosome biogenesis. The authors note a difference in the 5S rRNA, but this remains to be quantified (see note below). Altogether, these studies provide new insight into the mechanism of differences in skull development in a mouse model of FTSH, extending previous studies and providing a foundation for multiple areas of future research.

SUGGESTIONS TO AUTHORS

1. The authors have carefully responded to reviewer comments and softened the language surrounding ribosome biogenesis to prevent its over-interpretation. However, the analysis of small RNAs requires further consideration in both how the data was analyzed and its interpretation. Based on the data presented, the suggestion of disrupted ribosome biogenesis (line 320) is still not supported. For comparison of 5S rRNA, the exact same amount of total RNA should be loaded for each sample, which is not noted in the materials and methods. If this is indeed the case, please note this in the methods. To assess specific downregulation of 5S rRNA, the peaks should be quantified as a ratio (see Bir et al., 2024; PMID: 38192358 for an example of small RNA interpretation). As the data is currently represented in Fig. 6E, it appears that the mutant samples have less overall RNA as the pattern of the peaks does not appear to be significantly different across samples and all peaks are reduced in mutant samples. To be able to conclude a reduction in the 5S rRNA, this should be carefully quantified and statements in the discussion (lines 439-441) adjusted accordingly.

Author's reply: We thank the reviewer for these important suggestions and fully agree that the interpretation of the small RNA data requires careful consideration. To clarify, RNA extraction was performed identically for each skull and each Bioanalyzer run was conducted using RNA isolated from one skull per sample. The loaded RNA amount reflects the biological yield from each skull rather than a pre-normalized input concentration. This has been clarified in the method section in line 631.

Following the reviewer's suggestion and using the analysis approach demonstrated by Bir et al, we quantified the individual peak areas and calculated ratios for 5S rRNA and 5.8S rRNA relative to the total small RNA concentration. These additional metrics are included in Supplementary Table S2 and described in the method section (line 635-639). As the reviewer correctly noted, all small RNA peaks are reduced in mutant samples, indicating global depletion of small RNAs. Importantly, the internal peak ratios do not demonstrate a disproportionate reduction of the 5S region relative to other small RNAs. We have therefore adapted the section describing the Bioanalyzer data in the results and discussion (lines 295-298, 334-335 and 407-408). However, because this analysis is restricted to the small RNA fraction, it remains unclear whether 5S rRNA is reduced relative to the total RNA pool. Further studies are necessary to clarify the nature and extent of potentially impaired ribosome biogenesis, as noted in the discussion.

2. Y10b localizes to protrusions in WT but not mutant dura mater cells and then increased Y10b is observed in protrusions on mutant cultured NCCs. This seems to suggest that what might be happening in terms of Y10b and migration is different in NCCs versus dura mater cells, which could be an interesting area for future investigation.

Author's reply: We thank the reviewer for highlighting this interesting observation regarding Y10b localization differences between dura mater-derived cells and neural crest-derived cells and agree that this is a compelling direction for future research. We have incorporated this point into the manuscript to highlight this as an interesting direction for future investigation (page 16, line 417-420).

Minor points

1. The description of mutant osteoblasts treated with TGFB is missing from the section in lines 250-253.

Author's reply: Thank you for pointing this out. The description of mutant osteoblasts applies to both unstimulated and stimulated conditions. We have clarified the description now in line 233 and the figure legend.

2. Suggest rephrasing the statement regarding Fic 3C (line 197) to say that the staining in the PF suture is decreased in mutants as opposed to increased in WT

Author's reply: Thank you for the suggestion. We have changed the wording accordingly now in lines 183-184.

3. Fig. 1 legend (line 892) missing "6" when referring to the postnatal month

Author's reply: Thank you for noticing. The sentence has been adapted, now in line 851.

4. The image in S2A is helpful for addressing the areas imaged in Fig. 3C and Fig 4D and Fig. 7. To further improve the clarity, including a labeled dashed line to indicate plane of section for 3C and 4D (as well as 7E and 7F) would also be helpful. The sectioning of embryos should also be noted in the materials and methods section which is currently missing information on embryonic tissue harvesting and histology.

Author's reply: We thank the reviewer for the helpful suggestion. We have now included a schematic in Figure S2A to illustrate the orientation of the sagittal, coronal and transverse planes relative to the embryo for reference. All embryonic samples have been sectioned in the sagittal plane.

Information on embryonic tissue harvesting, processing and histology has been added in Materials and Methods (lines 485-491 and 496-497).

5. Similarly, indicating the plane of section in figure 5 would also be helpful.

Author's reply: All embryonic samples have been sectioned in the sagittal plane. This information was added to the Materials and Methods section and a schematic to illustrate has been added to Figure S2A.

6. Capitalize S in 5S line 444

Author's reply: We thank the reviewer for noticing. The S has been capitalized, now in line 411.

7. In the author contributions, the initials N.Q. are listed and do not appear to correspond with the listed authors.

Author's reply: We thank the reviewer for noticing. This has been modified.

Third decision letter

MS ID#: dev.204631R2

MS TITLE: The Sh3Pxd2bnee^{-/-} mouse: an attractive model unveiling novel developmental features in Frank-ter-Haar Syndrome

AUTHORS: Julika Huber, Siddharth Menon, Michael Lopez-Torres, Jason L. Guo and Michael T. Longaker

Dear Dr Huber,

I am happy to tell you that your manuscript has been accepted for publication in Development, pending our standard publication integrity checks.

Reviewer 1

Advance summary and potential significance to field

The authors have addressed all my concerns and I have no more questions.

Reviewer 2

Advance summary and potential significance to field

In this revision, the authors have thoughtfully replied to previous reviewer comments and addressed the concerns that were raised.

Comments for the author

I have no further suggestions.